# Feasibility of implementation of simplified management of young infants with possible serious bacterial infection when referral is not feasible in tribal areas of Pune district, Maharashtra, India

**Sudipto Roy**[1]\*, **Rutuja Patil**[1], **Aditi Apte**[1], **Kavita Thibe**[1], **Arun Dhongade**[1], **Bhagawan Pawar**[2], **Yasir Bin Nisar**[3], **Samira Aboubaker**[4], **Shamim Ahmad Qazi**[4], **Rajiv Bahl**[3], **Archana Patil**[5], **Sanjay Juvekar**[1], **Ashish Bavdekar**[1]

1 Vadu Rural Health Program, KEM Hospital Research Centre, Pune, Maharashtra, India, 2 Health Department, Zilla Parishad, Pune District, Govt of Maharashtra, India, 3 Department of Maternal, Newborn, Child and Adolescent Health and Ageing, World Health Organization, Geneva, Switzerland, 4 Department of Maternal, Newborn, Child and Adolescent Health, World Health Organization, Geneva (Retired), Switzerland, 5 State Family Welfare Bureau, Pune, Govt of Maharashtra, India

\* sudipto.roy@kemhrcvadu.org

**Data Availability Statement:** Data underlying the study's findings are available in the publicly

## Abstract

### Introduction

Neonatal infections are a common cause of death in India, but many families cannot access appropriate hospitals for its treatment due to various reasons. We implemented the World Health Organization PSBI management guideline when referral is not feasible within the public health system in Pune, India to evaluate feasibility, barriers and facilitators for its implementation.

### Methods

A national-level consultative meeting between government officials and study partners resulted in a consensus on adaptation and implementation in four demonstration sites in selected states in India. At the state and district levels, similar meetings to plan the implementation strategy and roles were held between KEM Hospital Research Centre (KEMHRC) Pune and the public health system Pune, Maharashtra. The public health system was responsible for implementation of the intervention at eight tribal primary health centres (PHC) in Pune district, India, including delivering the intervention and ensuring supplies of all commodities while KEMHRC was responsible for technical support including training of health workers, assistance in PSBI identification and management, data collection and documentation of the implementation strategy.

### Results

A total of 175 young infants with PSBI were identified and managed. Of these, 34 had critical illness (CI), 46 had clinical severe infection (CSI) and 10 were infants aged 0–6 days with

accessible INDEPTH Data repository at https://doi.org/10.7796/VADU.PSBI.2019.v1.

**Funding:** The study was supported by a grant from the Bill and Melinda Gates Foundation (BMGF Grant no. OPP1114815) to SAQ through the Department of Maternal, Newborn and Adolescent Health and Ageing, World Health Organization, Geneva, Switzerland. URL of funder website: www.gatesfoundation.org. KEMHRC Pune received funds through a Technical Services Agreement with WHO (WHO reference 2016/637074-0). SAQ, SA, YBN and RB from WHO were involved in study design, decision to publish and preparation of the manuscript.

**Competing interests:** The authors have declared that no competing interests exist.

fast breathing (FB) while 85 infants aged 7–59 days had fast breathing. Assuming a 10% incidence of PSBI among all live births, with 3071 live births recorded, the actual incidence of PSBI found in the study was 5.7%, resulting in an actual coverage was of 57%. Among the 90 infants with CI, CSI and FB in 0–6 days, who were advised referral to government tertiary care centre as per the PSBI guideline algorithm, 81 (90%) accepted referral while 9 (10%) refused and were offered treatment at primary health centres (PHC) with a seven-day course of injectable gentamicin and oral amoxicillin. All infants with FB in 7–59 days were offered treatment at PHCs as per the PSBI guideline algorithm with a seven-day course of oral amoxicillin. All except six infants who died and one with FB in 7–59 days, who was lost to follow-up, were successfully cured. Of the six who died, five had CSI and one had CI. Among the 81 infants with CI, CSI and FB in 0–6 days who accepted referral; 48(53%) were successfully referred to government tertiary facility while 33 (36.6%) preferred to visit a private tertiary health facility. The implementation strategy demonstrated a relatively high fidelity, acceptance and intervention penetration. Lack of training and confidence of the public health staff were major challenges faced, which were resolved to a large extent through supportive supervision and re-trainings.

## Conclusion

Management of PSBI is feasible to implement in out-patient facilities in the public health system, but technical support to the health system is required to jump-start the process. Fast breathing in 7–59 days old infants can be managed with oral amoxicillin without referral. A sustainable adoption of this intervention by the health system can lead to decrease in neonatal mortality and morbidity.

## Introduction

Globally 2.5 million neonatal deaths were documented in 2018, with Southern Asia having a neonatal mortality rate of 25 per 1000 live births [1]. Worldwide, neonatal sepsis, pneumonia and meningitis together result in up to 25% of all newborn deaths [2,3]. Highest incidence of neonatal sepsis (17,000/ 1,00,000 live births) was reported from India [4]. The mean incidence of bacterial infection among neonates in South Asia is reported as 13.2% per 1000 livebirths and bacterial infections accounted for 92% of the known causes of death due to possible serious bacterial infections. [5].

World Health Organization (WHO) labels potential neonatal sepsis as possible serious bacterial infections (PSBI) in young infants (0–59 days) and recommends referral to a hospital for injectable antibiotics and supportive care. However, in resource-limited settings, especially tribal and geographically difficult terrains, such referrals are often not feasible [6–9], due to distance, cost or cultural reasons [10,11]. Feasibility of treating severe infections in young infants in low resource settings on outpatient basis were first demonstrated in South-east Asia [6–9]. Subsequently, several large community based, randomized trials in Africa and Asia demonstrated the effectiveness of simplified antibiotic regimens in out-patient settings among young infants up to two months of age with signs of PSBI where referral was not feasible [12–16]. WHO in 2015 synthesized available evidence to develop global guidance for management of PSBI in young infants when referral is not feasible [17]. The Government of India also

approved a policy for management of PSBI where referral is not possible with simplified anti-biotic regimen by primary health care providers [18, 19].

However, implementation of such guideline in the public health system requires a multi-pronged approach and involves dialogue with the policy makers and program managers, understanding barriers and facilitators for implementation and providing technical support for implementation. While implementation challenges were well documented in controlled conditions in the above mention Asian and African countries, we could not find barriers to implementation of PSBI management within the public health system in India. Thus, an implementation research was planned across four different sites in India within the public health system to facilitate policy adoption and implementation of the WHO PSBI guideline. Maharashtra state has a relatively well functioning public health system and a private sector compared to many other states in India [20]. However, its tribal areas still have poor access to health care due to lack of proper roads, inadequate number of health facilities as well as low level of income and education among tribal population [21]. We describe here the process of implementation and outcomes of simplified management of PSBI in young infants in a tribal population from Western Maharashtra, India.

## Methods

### Aims

The aims of the implementation strategy were:

- To set up a demonstration site to deliver this intervention through policy dialogue and orientation meetings with Ministry of Health and other stakeholders at state and district level

- To demonstrate the feasibility of delivering simplified antibiotic regimens to young infants with PSBI where referral is not possible within programme setting through partnership between programme implementers and researchers

### Objectives

1. All first level health facilities will provide simplified outpatient management of PSBI when referral is not possible.

2. At least 80% of identified sick young infants will receive treatment.

3. At least 80% of the identified sick young infants will receive adequate treatment (Adequate treatment was considered as antibiotic doses for the first two days followed by at least 70% of doses over the next five days.)

### Design

The study was part of a multi-site and multi-country mixed-methods implementation research coordinated by WHO to test the feasibility of implementing PSBI guidelines where referral is not feasible in a programme setting. The other countries supported by WHO in this study were Democratic Republic of Congo, Ethiopia, Malawi, Nigeria and Pakistan. The Pune site study involved a partnership between King Edward Memorial Hospital Research Centre (KEMHRC) Pune, a research institution and the public health system in Maharashtra state and specifically, Pune district. A common protocol and common tools were developed by the

WHO team and used across four different sites in India. The guideline for management of PSBI in young infants, where referral is not feasible [18, 19, 22] constituted the study intervention while the implementation strategy included its introduction and implementation within the public health system in the study area. We have deliberately linked the intervention and the implementation strategy and not separated them while describing the study methods, as it reflects the actual 'real-world' conditions in which this study was conducted [23].

## The Implementation strategy

**Study partners.** The public health system, Pune district (implementation team), was responsible for implementing the study activities including identification and management of PSBI in young infants, ensuring effective referral systems and availability of medicines and equipment. The team from KEMHRC [Research and support team (RST)] was responsible for providing technical assistance, capacity development and supportive supervision to public health system workers as well as research data documentation. This clear division of responsibilities was a deliberate decision to ensure that the study could replicate programme conditions wherein the public health system is expected to implement the intervention with limited external assistance. Fig 1 shows the study logic framework for this study in the form of inputs, activities and outcomes by both study partners.

**Policy dialogue for implementing adapted guidelines.** A national level consultative meeting was held in 2016 to review, adopt and adapt the WHO guidelines for PSBI management which resulted in certain changes in the Indian IMNCI guideline (S1 Table). These changes included consolidation of signs and symptoms of PSBI based on which PSBI sub-classifications were modified and increased dose of oral amoxicillin was added. These changes were based on the evidence generated from the AFRINEST and SATT studies [13–16]. A policy decision was also made to adopt a regimen of seven-day injectable gentamicin and oral amoxicillin for PSBI when referral was not feasible [18,19]. The meeting resulted in a consensus on adaptation and implementation in four demonstration sites in selected states in India.

A second orientation and policy dialogue was conducted at the Maharashtra State level in early 2017 by study investigators from KEMHRC to ensure the buy-in from local health authorities and decision makers including Additional Director Health Services, State of Maharashtra (ADHS) and the District Health Officer (DHO), Pune. The public health system agreed to be a partner in the study and selection of the study area was decided at this meeting. This

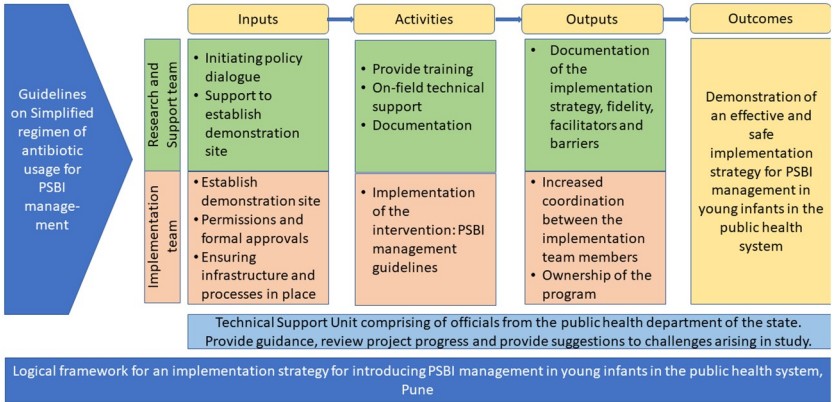

**Fig 1. Logical framework for an implementation strategy for introducing PSBI management in young infants in the public health system, Pune.**

dialogue resulted in the establishment of a Technical Support Unit (TSU). The TSU was an independent oversight body which comprised of senior public health officials from public health system, a paediatrician of government medical college and the investigators representing the RST. The role of the TSU was to provide guidance and leadership to the project and review the progress of the project and provide suggestions to challenges arising during the study conduct.

### Ethical considerations

The study was initiated after obtaining approval from KEMHRC and the WHO research ethics review committee. The study was implemented following good clinical practices and Indian Council of Medical Research guidelines for research involving human participants.

### Study area

The study was conducted across eight tribal Primary Health Centres (PHCs) situated in three blocks of Pune district namely Junnar, Ambegaon and Khed (Fig 2). All the selected villages in the study area were located in geographically difficult-to-access terrains but had public health infrastructure available for implementation. These PHCs are located in the Western Ghats of Maharashtra, covering total 1,40,000 population. Each PHC catered to around 20,000 population and comprised of 5–12 sub-centres (SC) serving about 3,000 population each. Details about the health infrastructure are presented in S2 Table and Fig 3.

### Baseline survey and mapping of referral system

A baseline survey was conducted by the RST in the eight selected PHCs and 15 randomly selected health sub-centres within these PHCs in order to asses readiness of facilities to implement the PSBI guidelines. This included review and availability of staff, knowledge, training and practice of auxiliary nurse midwives (ANMs), equipment, medicines, consumables and reporting systems relevant to PSBI management. The survey also included semi-structured interviews administered to 62 Accredited Social Health Activists (ASHA) and 121 mothers of young infants to understand their knowledge and practices related to care of young infants [Supplement]. Additionally, mapping was done of public as well as private health facilities

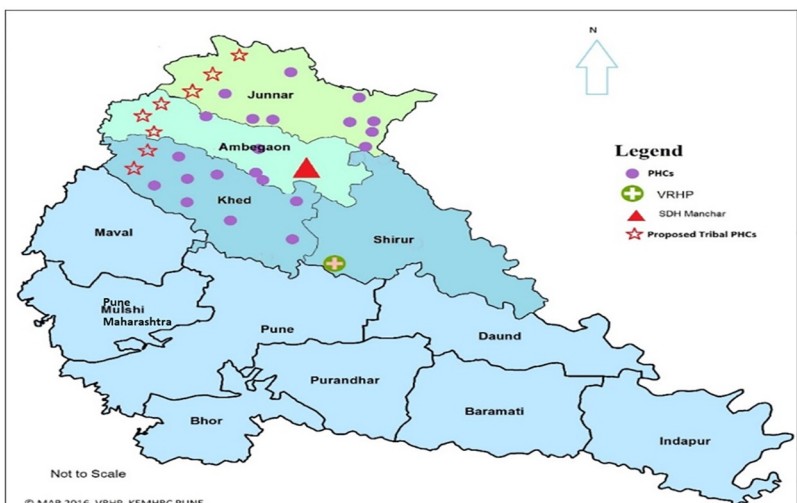

**Fig 2. Map of the study area in Pune district.**

Health Infrastructure of the study area

- Accredited Social Health Activists (ASHAs) are the community health workers who can identify signs and symptoms of illness in infants. ASHAs are essentially women who reside in the village that they cater to and should have completed secondary school education. ASHAs provide services to a population of 1000; services include routine home visits to pregnant women and newborn children. For their services they are provided an honorarium and target-based incentives. Trained to recognize danger signs in young infants.

- Auxiliary Nurse Midwife (ANM) at Sub-centres (SC): Caters to a population of 2000 to 3000. Provides OPD services; antenatal care; is skilled birth attendance trained for home and institutional deliveries; essential newborn care; post-natal visits to homes; provides immunization services. Manage PSBI cases under supervision of medical officer as per IMNCI guidelines.

- Primary Health Centre (PHC): Caters to a population of 20000 to 30000. The Medical officer (MO) is responsible for providing out-patient services, emergency care, antenatal care, management of normal uncomplicated deliveries, essential newborn care, including IMNCI strategy and immunization. Referral of PSBI cases after stabilization. In case referral not possible clinical management is provided as per guideline.

- Rural Hospital (RH): These are 30-bedded hospitals with in-patient facilities for uncomplicated cases. These do not have intensive care facilities.

- Sub-district hospital (SDH), Manchar: A 100-bedded facility without intensive care facilities; functions as a referral centre for lower government health facilties. Provides Specialized OPD services including pediatric care, emergency care and institutional deliveries including complicated deliveries, newborn care.

- Civil hospital, Pune: This is a district-level tertiary care hospital with neonatal intensive care facilities and is located in Pune district at an average distance of 80-140 kilometres from the study area. This was the primary referral centre in the study.

**Fig 3. Health infrastructure of the study area.**

with Neonatal Intensive Care Units (NICU)/ Special Newborn Care Units (SNCU) in increasing order of distance from the PHCs. The availability of ambulances in working conditions and the existing process of referrals were also documented.

## Trainings and orientation

Officials from the public health system and a member of the RST were trained as "master trainers" by trainers from WHO and IMNCI program of India in an extensive three-day training on assessment, classification and management of PSBI along with newborn care. The training included interactive class room training as well as hands on clinical practice. The algorithm for identification and management of PSBI was prepared from adopting updates from the WHO PSBI guideline on managing PSBI in young infants when referral is not feasible [17] to the WHO young infant IMCI chart booklet, 2019 [22]. Although the WHO young infant IMCI chart booklet was printed in 2019, the draft was used to adapt the Indian guideline to manage PSBI when referral was not feasible. The master trainers then used the same training methods and materials to train all medical officers (MO) and ANMs in the study area. The MOs with support from RST, oriented ASHAs on identifying danger signs of PSBI and essential newborn care, at their respective PHCs. Twelve MOs, 68 ANMs and 265 ASHAs in the eight selected PHCs were trained through these sessions.

## The Intervention

### Implementation of simplified management of sick young infants (Fig 4)

   **i. Identification of sick young infants.**   The ANMs, as part of their routine activities recorded all pregnancies and births in their coverage area and ASHAs visited all neonates on days 1, 3, 7, 14, 30 and 42 of birth as per national guidelines. During these visits, ASHAs counselled the caregivers about danger signs of PSBI in addition to providing newborn care and identifying danger signs of illness if they came across a sick young infant. Young infants with axillary temperature 37.5˚C or above, axillary temperature less than 35.5˚C, fast breathing, chest indrawing, convulsions, difficulty in feeding or reduced movements were then brought to ANMs or MOs for further assessment and intervention. At PHC/SC level, the sick young infants were assessed by MOs or ANMs for signs of PSBI and referred to Civil hospital, the

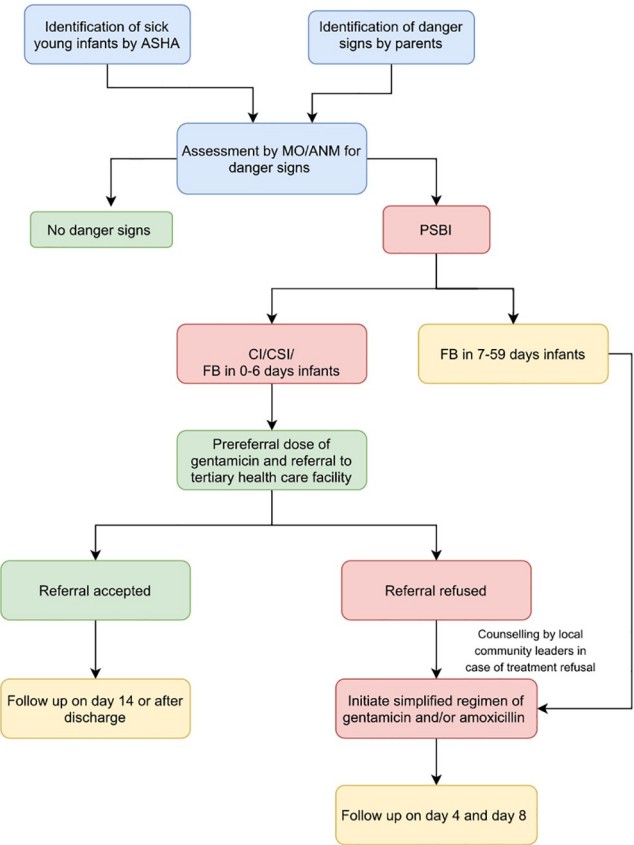

**Fig 4. Flow chart of PSBI identification and management.**

tertiary care centre in our study, if needed. The sick young infants were classified as either PSBI, with sub-classifications of clinical severe infection (CSI), critical illness (CI) and fast breathing (FB) in 0–6 days old infants, or FB in 7–59 days old infants.

**ii. Referral to tertiary health care facilities.** As per guidelines, caregivers of young infants classified as CSI, CI or FB (only in 0–6 days old) were counselled about the need for hospitalization and were advised referral by MOs and ANMs to the civil Hospital the for treatment. The rural hospitals and sub-district hospitals only manage uncomplicated infection cases in infants, provide immediate post-natal care but do not have neonatal intensive care facilities to manage serious illness. Hence these were not considered as referral facilities for PSBI cases. In many cases, ASHAs accompanied the young infants to the tertiary care. Such young infants were given first dose of gentamicin intramuscularly before referral to tertiary health care facilities. For every referral, a referral slip was handed over to the parent to show to the tertiary care facility. The caregivers were given instructions to maintain warmth and provide breastfeeding to the baby during transportation. In infants classified as CI, CSI or FB (only in 0–6 days old), whenever the referral was not feasible, MO/ANM continued to provide care as per adapted PSBI management algorithm [22].

**iii. Treatment when referral is not feasible or referral is not recommended.** When referral was not feasible, even after making all efforts to counsel and refer, infants aged 0–6 days with FB and young infants with any sign of CSI were treated with a course of intramuscular gentamicin 5–7.5 mg/kg once daily for 7 days and oral amoxicillin 50mg/kg twice daily for 7 days on an outpatient basis. Infants with CI were started on a course of intramuscular

ampicillin and intramuscular gentamicin daily for seven days. The MOs had overall responsibility for identification of PSBI, choosing the appropriate treatment and administering the treatment. All doses of injection gentamicin and first daily dose of oral amoxicillin were administered by the MO or ANM in consultation with the MO at the PHC and the second oral amoxicillin dose for the day was administered by mother after being taught to ascertain the correct dose and method of administration by ANM. All young infants aged 7–59 days with FB were treated with oral amoxicillin 50mg/kg twice daily for 7 days as per the treatment algorithm [22]. Caregivers were taught to administer oral amoxicillin at home, counselled on identifying signs of deterioration and were asked to bring the infant back to the ANM if there was any such sign. If the child's health would deteriorate, they were again advised referral. The protocol defined adequate treatment as antibiotic doses for the first two days followed by at least 70% of doses over the next five days.

**iv. Monitoring and follow up.** Young infants with any sign of CSI or CI, or FB (only in 0–6 days old infants) who received outpatient treatment were followed up every day till day 7 of treatment. Infants aged 7–59 days with only FB who received oral amoxicillin were followed up on the 4th and 8th days of starting treatment by ASHA/ANMs to assess the infant's condition and adherence to treatment, while if they were treated at private health facilities, follow-up was done on the 8th day only. In case of referred sick young infants, first telephonic follow up was done by ANM during hospitalisation and second follow up was done at home by ANM with ASHA after hospital discharge within 14 days of onset. All clinical assessments, identification and management of PSBI by ANMs and MOs were recorded in IMNCI forms at PHCs (Fig 4).

## Activities of the KEMHRC RST

The RST, in consultation with the Implementation team developed the study protocol and study tools. It organized and conducted training for all health workers in the study area. The RST did not participate in any implementation activities but was available for on-field support to the health care workers and whenever needed provided additional training related to assessment, classification and management of PSBI cases. Study activities were jointly supervised by members of the RST and senior district level public health authorities. RST members were consulted by MOs/ANMs regarding identification of PSBI and use of the treatment algorithms for appropriate management, in approximately 10% of all PSBI cases managed. The consultations were done as and when required and there was no pre-decided pattern. The RST coordinated with the block and district level authorities to solve challenges and problems in the field. The RST transcribed data related to PSBI case management from records maintained by healthcare workers onto case record forms developed for this study. Finally, the RST was responsible for analysis and presentation of results from this study.

## Communication

Based on the results of the baseline survey, information, education and communication tools were developed by the RST with inputs from the health workers. This consisted of charts and booklets depicting signs and symptoms of PSBI and their management as per adapted study guidelines when the referral is not feasible. The RST communicated with the Implementation team during their weekly meetings at the respective PHCs in addition to the frequent face to face and telephonic interactions during and after identification of the suspected PSBI cases. The RST and Implementation team also used multimedia technology like WhatsApp videos to confirm the signs in five to six young infants with suspected PSBI. In addition, there were regular meetings with block and district level authorities to discuss the challenges faced and devise

solutions to overcome these. These meetings were held once every month for the initial five-six months as most of the challenges were identified during these initial months into the study and thereafter, once every three months. These helped in devising solutions to improve case identification, ensuring uninterrupted supplies of medicines and consumables and ensuring referral mechanisms could function appropriately.

## Data management and analysis

The data were entered into electronic record forms created using REDcap software. The data management was done on local standalone server with back up facilities. Anonymized dataset was extracted and was used for analysis for quality control. Descriptive analysis was done using Microsoft Excel calculations of frequencies and proportions.

## Study period

The baseline and preparatory activities were conducted between March 2017 and July 2017 while field activities were conducted between August 2017 and March 2019.

## Results

### Preparatory phase

**Baseline survey.** The baseline survey of the facilities in the study area showed that all the eight PHCs and 64 SCs in the study area were functional in terms of availability of human resources, medicines and supplies, updated documentation and patient flow. There were three 30-bedded rural hospitals (one in each block) located within 20–40 km distance from each PHC whereas a 100-bedded subdistrict hospital was located at 70–80 km from the PHCs (Fig 3). There were approximately 8–10 private health facilities within the study area, with 2–3 having intensive care units. All PHCs were equipped with one ambulance each for referral to a higher health facility. However, vehicles were not available at all times and patients sometimes needed to spend money for fuel or arrange for alternate transport to reach a higher health facility. There were adequate human resources working at the eight PHCs. The details of the staff and population catered for the eight PHCs in given in S2 Table.

We found that 95% ASHAs reportedly conducted home visits post-delivery as per government guidelines and about 80–85% ASHAs were aware about overall care of young infants and counselling for newborn care. Amongst the caregivers, 90% mothers had received counselling on care of young infants post-delivery. However, only 55–60% ASHAs on an average, were aware of danger signs in young infants and only 50% of the mothers on average received counselling on the same. MOs and ANMs had been trained in IMNCI. Before initiation of this project, none of the ANMs and MOs reported to have identified and managed PSBI in young infants in the past three months. Sick young infants approaching the PHCs were routinely referred to higher-level health facilities by the MOs. A crucial reason cited was lack of confidence and fear of mishap in handling sick young infants at a peripheral health facility. However, they showed readiness to manage cases with PSBI if they were provided with appropriate training and technical support. This survey also revealed that caregivers of young infants from the tribal population generally did not show readiness for accepting referral and preferred home-based care due to cultural and socioeconomic reasons (lack of belief in healthcare, non-affordability, unavailability of transport, loss of wages). Most of the ANMs were not experienced in administration of intramuscular gentamicin in infants. Most of the PHCs were equipped with baby warmer but were not adequately equipped with weighing scales, thermometers and stopwatches or timers to count respiratory rate. Oral amoxicillin was not

available at all centres at the time of the survey, whereas injectable gentamicin vials in a dose of 80mg/2ml and syringes for intramuscular administration were available.

**Discussion at State and district level for assurance of supplies.** Observations from the baseline assessment were communicated by RST to the senior officials from the public health system in face to face meetings. Assurance for support was obtained from the government officials in order to maintain the health staff and adequate supplies of amoxicillin, gentamicin and consumables. Before implementation, the PHCs and SCs were strengthened by the RST with provision of weighing scales, thermometers and stopwatches/timers for counting respiratory rate to all ASHAs.

## Implementation phase

**Birth surveillance and postnatal visits.** During the study period, total 3071 live births were recorded from all the institutions. Of these, 2001 (65%) could be followed up for post-natal home-based care by ASHAs. Post-natal home visits were made by ASHAs in majority of the cases (S3 Table). A decrease from the first post-natal visit in subsequent postnatal visits were due to parents moving out of study area after discharge from the health facility and lack of documentation by the ASHAs for some of the visits.

**Identification and management of sick young infants.** A total 545 sick young infants were assessed during the study implementation phase, of which 175 had signs of PSBI (34 CI, 46 CSI, 10 FB in 0–6 days old infants and 85 infants aged 7–59 days with FB). (Table 1). Assuming a 10% incidence (5, 13, 14, 15, 16) of PSBI among all live births, with 3071 live births recorded, the expected number of young infants with any sign of PSBI would be 307. We identified and treated 175 young infants with PSBI indicating actual coverage of 57%.

**Identification of illness.** About 38.3% PSBI cases were identified by the families, 35.4% by ASHAs and the rest were identified by MO/ANMs.

**Referral to government or private tertiary care and management of sick young infant at PHC/SC (Table 2).** Among the 90 PSBI cases that were eligible for referral and were advised referral to a government tertiary care centre, 81 (90%) accepted referral. Among 81 young infants with any sign of PSBI who accepted referral, 48 (59.3%) were hospitalized at a government tertiary care, while 33 (40.7%) preferred to visit a private facility. Pre-referral doses of gentamicin were given in only 7.1% (6 out of 84) of the cases. Those who refused referral (n = 9) received simplified antibiotic treatment on outpatient basis at PHC/SC level, of which five were classified as CSI, three as fast breathing in infants aged 0–6 days and one had CI. All 85 cases of FB infants aged 7–59 days were offered treatment at PHCs, 62 (72.9%) infants accepted and were treated with oral amoxicillin, while 22 (25.9%) infants refused treatment and went to a private facility. One case of FB in 7–59 days old infant, refused treatment with oral amoxicillin as the parents immediately migrated out of the study area; this case could not

**Table 1. Identification of sick young infants during study duration (August 2017 to March 2019) in Pune.**

| Parameters | All sick young infants (N = 175) | Critical illness (CI) (N = 34) | Clinical severe infection (CSI) (N = 46) | Fast breathing (FB) in 0–6 days (N = 10) | Fast breathing (FB) in 7–59 days (N = 85) |
|---|---|---|---|---|---|
| Brought by families to PHC/SC, n (%) | 67 (38.3) | 5 (14.7) | 20 (43.5) | 4 (40.0) | 38 (44.7) |
| Identified by ASHAs in the community and referred to PHC/SC, n (%) | 62 (35.4) | 11 (32.3) | 15 (32.6) | 1 (10.0) | 35 (41.2) |
| Identified at PHC/SC by ANM/MO following birth or during immunisation sessions, n (%) | 46 (26.3) | 18 (52.9) | 11 (23.9) | 5 (50.0) | 12 (14.1) |

PHC- Primary health centre, SC- subcentre, ANM- Auxiliary nurse midwife, MO- medical officer,

**Table 2. Referral, treatment and outcome of sick young infants during study duration.**

| Parameters | All infection cases (N = 175) | Critical illness (CI) (N = 34) | Clinical severe infection (N = 46) | Fast breathing in 0–6 days (N = 10) | Fast breathing in 7–59 days (N = 85) |
|---|---|---|---|---|---|
| **Referral and treatment of sick young infants, n (%)** | | | | | |
| Referral to tertiary government facility accepted | 48 (27.4) | 27 (79.4) | 17 (37.0) | 4(40.0) | |
| Referral was accepted but went to private facility | 55 (31.4) | 06 (17.6) | 24 (52.2) | 3(30.0) | |
| Referral not possible and treated at PHC/SC | 9 (5.1) | 1 (2.9) | 5 (10.9) | 3(30.0) | |
| Referral not recommended and treated at PHC/SC | 62(35.4) | - - - | - - - | - - - | |
| Refused referral and also refused treatment at PHC/SC | - - - | | - - - | - - - | |
| **Follow up of all sick young infants, n (%)** | | | | | |
| Completion of at least one follow-up visit by ANM [#] | 174 (99.4) | 34 (100.0) | 46 (100.0) | 10(100.0) | 84 (98.8) |
| **Final treatment outcomes, n (%)** | | | | | |
| Clinical treatment success[$] | 168 (96.0) | 33 (97.1) | 41 (89.1) | 10 (100.0) | 84 (98.8) |
| Death | 6 (3.4) | 1 (2.9) | 5 (10.9) | 0 (0.0) | 0 (0.0) |
| Outcome unknown | 1 (0.6) | 0 (0.0) | 0 (0.0) | 0 (0.0) | 1 (1.2) |

[#]—follow up was completed within 8 days for sick young infants treated at PHC and telephonic follow up for infants referred and hospitalised at tertiary centre/private facility followed by home visits by ANM after discharge; WHO guideline recommends Day 4 mandatory follow-up; [$]—The treatments include treatments given at the tertiary referral as well as simplified regimens offered at PHC/SC.

be followed up for outcome. We could not collect treatment information on treatment given from private health facilities.

**Clinical outcomes in sick young infants (Table 2).** Of 175 infants, 168 (96%) were cured of their illness, with similar cure rates at public and private healthcare centres. Six (3.43%) young infants with PSBI (five CSI and one CI) died. All these infants were very low birth weight (average birth weight 1.4 ± 0.21 kg) and pre-term (less than 34 weeks gestational age) and all were diagnosed with signs of PSBI on day 0 or 1 of life. All the six infants were referred to the district hospital but only four accepted the referral (including one self-referral to private health care). Two infants refused referral and were started treatment at the PHC on day-1 of identification. One received two doses of oral amoxicillin and no injection gentamicin and the other received three days treatment with injection gentamicin and oral amoxicillin. Both were discharged against medical advice and subsequent deaths (on day three and six respectively) were reported by ASHAs following home visits. Among infants aged 7–59 days with FB, treatment success was 100%, irrespective of the place of treatment.

**Treatments given to sick young infants when referral was not recommended or possible (Table 3).** Nine sick infants with PSBI (5 CSI, 1 CI, 3 FB in 0–6 days old infants) for whom referral was not possible, received simplified regimen at PHC/SC. Six (67%) of these cases received complete treatment with gentamicin and oral amoxicillin, one sick young infant with CSI received three injections of gentamicin and three days of oral amoxicillin and one received two doses of oral amoxicillin and no injection gentamicin. One infant did not receive any treatment as its weight was 1.4 kg and MOs/ANMs of that particular PHC refused to treat such very low birth weight babies. Since its parents refused referral, the infant was offered Kangaroo mother care and closely followed up and went on to recover from the illness. This infant had

**Table 3. Treatment for sick young infants when referral was not recommended or feasible.**

| Parameters | All PSBI cases (N = 9) | Critical illness (CI) (N = 1) | Clinical severe infection (N = 5) | Fast breathing in 0–6 days (n = 3) | Fast breathing in 7–59 days (n = 62) |
|---|---|---|---|---|---|
| Intramuscular gentamicin doses (5–7.5 mg/kg once daily), n (%) | | | | | |
| Received all 7 injections | 6 (66.7) | 1 (100.0) | 2 (40.00) | 3(100.0) | |
| Received 3 injections | 1 (11.1) | 0 (0.0) | 1 (20.0) | 0(0.0) | |
| Received 1 injections | 0(0) | 0(0.0) | 0 (0.0) | 0 (0.0) | |
| Received no injection[#] | 2(22.2) | - - | 2(40.0) | - - | |
| Oral Amoxicillin doses (50mg/kg twice daily), n (%) | | | | | |
| Received all 14 doses of treatment | 61(84.7) | 1(100.0) | 2(40.0) | 3(100.0) | 55(88.7) |
| Received 10–13 doses of treatment | 5 (6.9) | 00(0.0) | 0(0.0) | 0(0.0) | 5(8) |
| Received 6–9 doses of treatment | 1 (1.4) | 0(0.0) | 0(0.0) | 0(0.0) | 1(1.61) |
| Received ≤5 doses of treatment | 4 (5.6) | 0(0.0) | 2(40.0) | 0(0.0) | 1(1.61) |
| Received no amoxicillin[#] | 1 (1.4) | - - | 1(20.0) | - - | 0(0.0) |

The protocol defined adequate treatment as antibiotic doses for the first two days followed by at least 70% of doses over the next five days.

[#] The infant did not receive any treatment as body weight was less than 1.5 kg;

hypothermia and movements were lesser than normal. Breastfeeding was initiated and continued adequately.

Of the 85 cases of FB in 7–59 days infants, 62 infants aged 7–59 days with FB pneumonia received treatment at PHC/SC. Of these, 55 (88.7%) completed treatment with oral amoxicillin and 7 (11.2%) infants received less than adequate treatment for 6–13 days. The remaining 22 infants who took treatment from private facilities were followed up on the 8th day of the illness and were found to be cured. One case was lost to follow-up.

## Study progress and facilitators and challenges for implementation

The number of cases managed in the first five months were less than expected, with few health staff identifying and managing PSBI cases. The RST provided on-field support to all health staff while public health officials, through their feedback mechanisms, encouraged health staff to identify and manage PSBI cases. With these efforts, case reporting did increase over the remaining 15 months of the study period (Fig 5).

**Facilitators for success.** The ownership of the study by the public health officials at the state and district levels and establishment of TSU were major facilitators that contributed to a smooth implementation of the study. This buy-in of the public health system was a crucial aspect of this implementation research study in terms of potential sustainability and scale-up.

Training sessions for all health staff in the study areas played a major role in enhancing the knowledge and capacity building of MOs, ANMs and ASHAs in identification and management of the PSBI cases. This also provided opportunity for trainers to allay their apprehension and lack of confidence about handling sick young infants. Study-related communication at all levels in public health system increased awareness about the PSBI and importance of treatment and timely referral in these cases among the Implementation team. This helped the Implementation team in prioritising management of sick young infants through simplified regimes and increased their ownership towards the program. These was evident from the change in

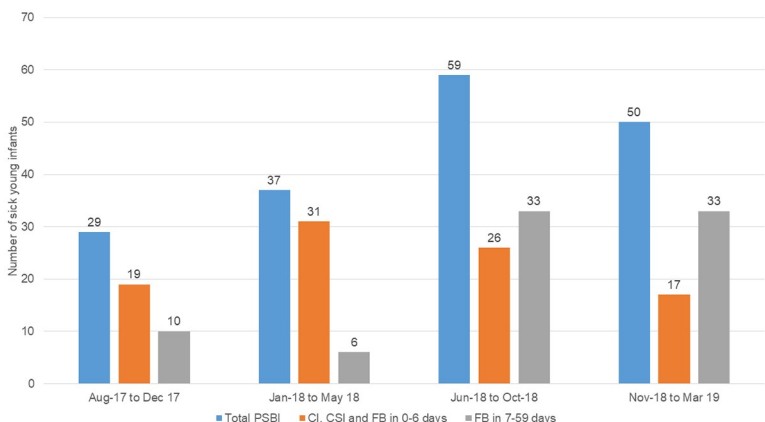

**Fig 5. Progression of identification of sick young infants with PSBI.**

practices regarding managing PSBI in young infants, with the baseline survey showing that no PSBI case was managed at PHCs prior to the study implementation.

A robust and effective referral system ensured that a majority [81 out of 90 (90%)] of young infants with PSBI, who were eligible for referral to government tertiary care health centres, actually accepted referral to such centres. The role of ASHAs was specifically important in persuading the caregivers to accept the referrals. In some cases, ASHAs, being the member of the same community, with help from the local community leaders and village heads also arranged for logistics or monetary support from within the village for transport of the sick young infants to health care facility when ambulance was not available immediately.

**Challenges faced.** Our study faced various challenges and barriers during implementation, which resulted in numerous modifications of the predefined protocol (Table 4).

## Discussion

Broadly speaking, this implementation research shows that it is feasible to implement the strategy of early identification, timely referral and simplified management of sick young infants in the public health system in tribal areas of Maharashtra. The public health system has successful managed 90% of the PSBI cases identified during the study period independently while 10% required consultation with the RST implying the study was implemented as planned.

We had reasonable treatment coverage of approximately 60% in our study. Similar implementation research studies on PSBI in young infants from African and Asian countries reported variable treatment coverages of 16.3% in Sylhet and Chittagong in Bangladesh [24], 50% in Ethiopia [25], 42% in Kushtia district, Bangladesh [26], 63.8% in Malawi [27] and 95% in Nigeria [28]. Thus, except for the Nigeria study, our coverage proportions were comparable with other reported coverages. However, our coverage could have been higher; a major factor in our study area was the preference of private health facilities which might explain where the remaining estimated 40% cases of PSBI would have been managed. Another factor could be that some PSBI cases visited the sub-district and district hospitals directly. However, we did not collect data from these hospitals as this was outside the study protocol. We estimate that the actual coverage may need a correction factor. The coverage achieved in our study was a result of effective and timely coordination between different facilities and cadre of the public health system. Ownership of the program and supervision by senior public health officials were crucial for achieving these coverage proportions. Along with it, support from the KEMHRC RST, especially in the early phase of the study played an important role in this. The

**Table 4. Challenges face during implementation research, steps taken and end result.**

| No | Issue | Steps taken | End result |
|---|---|---|---|
| 1 | Inadequate case reporting in the first six months and lack of coordination between ASHAs and ANMs | Re orientation of ANM and ASHAs in identifying PSBI | Increased case reporting in next 15 months period of the study. However, few ASHAs and ANMs still reported fewer cases/population than others |
| | | Counselling of ANMs by MOs to increase case-finding efforts | |
| | | On-field support for low-performing ASHAs and ANMs | |
| 2 | Documentation by ANMs/MOs | Re-orientation sessions and hand-holding for recording complete data for individual cases. | Most ANMs recorded adequately; ANMs in busier PHCs complained of lack of adequate time to fill in IMNCI forms completely |
| 3 | Parents taking their sick Young infants to private providers in-spite of advice for referral to government tertiary care centres. Primary reasons given for private preference were relatively more confidence in and relative ease of negotiating private health-care processes | MOs and ANMs counselled parents for referral to government tertiary care centers | Referrals to Government facilities improved over the study period, but few parents still preferred private providers |
| 4 | A smaller proportion of PSBI cases received pre-referral doses. | MOs/ANMs supported in administering pre-referral Gentamicin injections | The number improved, but only 6 PSBI cases received pre-referral doses in the study. |
| 5 | Unavailability of ambulance service needed for prompt transport of sick young infant to referral centre | ASHAs and ANMs coordinated with the village heads for making a vehicle available for the patient. | The sick young infant got access to treatment at tertiary care centre. |
| | | District health authorities ensured availability of ambulance at all PHCs. | |
| 6 | Unavailability of antibiotics in some PHC/SCs in the first three months of the study. | The RST coordinated between different health facilities to make the supplies available to the ones which did not have enough supply. District health authorities ensured availability of medicines at all PHCs | The antibiotics were subsequently made available through the study period |
| 7 | Lack of experience in administration of gentamicin injection in low birth weight infants in the initial months of the study. | Retraining and confidence building was done by the RST. MO/ANMs were trained in administering injections even to VLBW babies as part of routine immunizations | The MO/ANM did not administer the prereferral dose in spite of the trainings as they were afraid of being responsible in case of mishap |

processes conducted in our study can again inform the successful adoption of these guidelines in other districts and states in India, subject to local modifications.

One of the most important outcomes of this study was that fast breathing pneumonia in 7–59 days young infants was effectively and safely treated with oral amoxicillin on an out-patient basis without referral in a programme setting. Almost three-fourth of such infants were successfully treated by MOs/ANMs and were completely cured, with most receiving adequate doses of oral amoxicillin. The rest preferred private healthcare centres where they could have received treatment with other antibiotics and were also completely cured. MO/ANMs were able to ascertain preference for private facilities only after these cases came to SC/PHC and were diagnosed with PSBI. There was no scope in the study for separate data collection from those who directly went to private facilities. Primary reasons given for private preference were relatively more confidence in and relative ease of negotiating private health-care processes. In fact, this finding from the study will help to inform policy regarding PSBI management when referral is not feasible to a larger public sector hospital, wherein greater public-private partnership in health can be a recommendation to reduce infant mortality. However, from studies on antibiotic prescriptions, it has been seen that very few private facilities treat infections in children with simplified antibiotics such as gentamicin and amoxicillin, with most preferring higher-level antibiotics [29, 30]. In addition, affordability of such facilities remains a challenge, especially for the tribal communities as in the study area. Thus, a balanced view about the appropriateness, affordability as well as accessibility of private health facilities needs to be considered. In any case, this finding enables the public health system to effectively triage such cases and avoid referring them unnecessarily to tertiary care centres, thus reducing

the workload at these centres as well as reducing the exposure of these young infants to higher-level anti-microbials. In the Malawi study, out of 150 young infants with FB, cure rate was achieved in 96% infants [27]. The study in Kushtia, Bangladesh reported that among 475 young infants with FB, approximately 80% of caregivers reported that their infant had received oral amoxicillin for five days and 74% for seven days and the treatment failure rate was approximately 5% [26]. Similar proportions were reported in the Nigeria study with All 7–59 days old young infants with fast breathing who received treatment at the outpatient improved with oral amoxicillin [28]. Thus, our study results had comparative results for outcomes of FB in 7–59 days infants, though, our numbers were lesser. Thus, it is feasible to enact this recommendation within the public health system in India.

The current Government of India guideline [18] recommends treatment by injectable gentamicin and oral amoxicillin for fast-breathing pneumonia in young infants aged 7–59 days. Our study data can help to inform the guidelines for necessary revision.

A relatively low case fatality rate of 3.4% (six deaths out of 175 cases) showed that the adapted PSBI guidelines can help reduce neonatal and infant mortality, if implemented appropriately by the public health system. However, considering the relatively low number of PSBI cases, we need to interpret this fatality rate with caution when applying to other areas as well as while scaling-up the intervention.

A striking finding from our study was the high acceptance (90%) of referral to tertiary care centres among eligible infants with PSBI. While all eligible PSBI infants were advised referral to government tertiary care centres, a substantial proportion preferred to visit a private tertiary care. The high number of referrals to government tertiary care hospitals was due to presence of effective communication amongst the care-providers and increased awareness and an effective referral system. Further, linkages with the tertiary government healthcare centres, facilitated through senior state level health officials, helped in successful referrals. While not assessed in details, on interviewing some parents, we found that the major reason for going to private facilities was easy acceptability, familiarity with the private physicians and proximity of the private facility as compared to a public facility. We thus recommend encouraging partnership between the public and private sector for healthcare delivery and implementing the guidelines for PSBI management, considering that one third of sick infants were managed at the private healthcare facilities were perceived to be more accessible to the general population than government tertiary care unit. This relates to our study area where private health facilities are available. Healthcare-seeking is a choice and it was clear from our study that a sizeable proportion preferred private facilities. Thus, in a population where there is some preference for private health facilities, it would be beneficial for the population if a private-public partnership with common treatment protocols is built, as there would be a choice for seeking care. This is applicable for many parts of India, wherein private facilities account for 60% of all outpatient visits and 40% of all inpatient admissions. However, quality, cost and completeness of treatment in private care remain a challenge. This aspect needs further investigation and we agree that the private health sector, where available, can play a valuable role in managing sick young infants. Our referral refusal proportion differs considerably from other similar studies, with referral refusal as high as 83% in Bangladesh, 90% in Malawi and 97% in Nigeria {24, 27, 28]. The high acceptance of referral in our study could limit generalizability of this study to settings where such high referral is not possible.

Only about 10% of the PSBI cases who were recommended referral, refused referral and were administered simplified management at PHC/SC level. The common reasons for refusal of referral were lack of manpower and family support, financial issues and lack of transportation of facilities. We need to further explore ways to facilitate referrals for such cases, since the recommended guidelines for PSBI cases is treatment at a tertiary care facility. The quality of

treatment received at PHC level was appropriate for two-thirds of the CSI, CI and FB in 0–7 days PSBI cases in those who accepted the treatment. While this was based on only 9 cases for which referral was not possible, of which quality treatment was given to 6 cases, we admit that even for such a small number, the quality should have been higher and is a limitation in our study. The one infant with a birth weight of 1.4 kg and not given any treatment, had hypothermia, which could be environmental and not due to PSBI. This infant was feeding well and was closely monitored till recovery.

An important finding from this study was the relative safety of the interventions as well as the implementation strategy, with none of the patients reporting adverse events. However, one limitation in our study is that we can only comment about safety of amoxicillin in FB in 7–59 days infants with confidence.

The most crucial factor for successful implementation was the complete ownership by the public health system, especially at the state level. The involvement of senior leadership in the public health system also ensures that the intervention can be scaled-up to other parts of the district and the state and can possibly be sustained in the form of a programme, though further policy-related dialogues are needed for this. Secondly, technical support to the public health system for an initial period of time helps to resolve many issues and streamline the implementation process.

## Readiness of the public health system

We found that all health staff, including MOs, ANMs and ASHAs needed re-training in IMNCI including PSBI management. There were no major challenges regarding availability of medicines and consumables, however, equipment such as accurate weighing machines, stopwatches/respiratory rate timers needed for managing PSBI was inadequate. The equipment was made available after informing the district authorities. These issues, though can be relatively easily resolved within the public health system, by timely and appropriate communication need to be assessed before commencing implementation.

## Acceptability and implementation penetration

Within the public health system, the study strategy was generally well accepted. However, our study showed that initially, facilitation and hand-holding for a limited time, built confidence of entire public health system; especially the primary healthcare workers. This was reflected in an increase in the number of cases identified and managed after as the study progressed. However, the hesitancy of ANMs to administer pre-referral injectable gentamicin in young infants due to fear of adverse effects could not be overcome sufficiently in spite of refresher training and constant on-field support. The awareness about management of sick young infants and feeling of responsibility increased amongst Implementation Team, due to which the MOs/ANMs who did not manage a single sick infant with PSBI at baseline, could effectively manage sick infants at PHC/SCs when referral was not feasible. The acceptance amongst public health system and top-down communication was found to be a major facilitator.

## Fidelity and system adaptations required

In PSBI cases, with 90% young infants successfully referred, though only a little over half accepted referral to government tertiary facilities as recommended, we can say that fidelity, in terms of referral to government tertiary facilities, was not achieved as expected. However, this was probably driven by the prevalent care-seeking preferences in the study population and needs to be explored further., while, among those not referred, further efforts need to be taken to ensure that all such infants are adequately treated on an out-patient basis. These efforts

could include more effective counselling of care-givers, greater involvement of local community leaders in counselling efforts and more support from senior public health officials. No major system adaptations were required. However, a limitation of this study was the low proportion of pre-referral injectable gentamicin in this study indicating that there is a need for motivation and confidence-building, including support of senior public health authorities, of MOs, ANMs and ASHAs to administer pre-referral gentamicin.

This implementation science research was appreciated by the public health system and in future, the RST was invited to continue working with the public health system. The TSU and the public health system is keen to scale up these adapted PSBI guidelines to entire state of Maharashtra. This will be done by scaling up the training activities and awareness program through the district training teams in the state. However, in order to make the program sustainable, it is important that the role of RST is taken over by the public health system and TSU.

## Conclusion

The study demonstrated that the identification and management of PSBI in young infants can be implemented at out-patient facilities in the public health system. Fast breathing in 7–59 days old can be effectively managed with oral antibiotic on an outpatient basis without referral. Technical support to the health system is required to jump-start the process; however once started, a sustainable adoption of this intervention by the health system can lead to decrease in mortality and morbidity in young infants.

## Supporting information

**S1 Table.**
(DOCX)

**S2 Table.**
(DOCX)

**S3 Table.**
(DOCX)

**S4 Table.**
(DOCX)

**S1 Questionnaire.**
(PDF)

**S2 Questionnaire.**
(PDF)

## Acknowledgments

- Dr. Makarand Ghorpade, study coordinator, KEM Hospital Research Centre Pune for helping to coordinate study activities

- Technical Support Unit members including Dr. Arti Kinikar, Dr. Sanjeevkumar Jadhav and Dr. Ashish Bharti for providing technical overview

- Taluka Health Officers of Junnar, Ambegaon and Khed blocks, Health Department, Zilla Parishad, Pune for supervising PHC staff

- MOs, ANMs and ASHAs of all eight study PHCs, Health Department, Zilla Parishad, Pune for implementing PSBI guidelines

- Dr. Harish Chellani; Dr. Sugandha Arya, Indian national IMNCI master trainers for conducting the study training

- Field Research Assistants and clinical coordinators of KEMHRC team for supporting the PHC staff

- Kaushik Ghosh, developer of data management system in Redcap

## Author Contributions

**Conceptualization:** Sudipto Roy, Yasir Bin Nisar, Samira Aboubaker, Shamim Ahmad Qazi, Rajiv Bahl, Sanjay Juvekar, Ashish Bavdekar.

**Data curation:** Sudipto Roy, Rutuja Patil, Aditi Apte.

**Formal analysis:** Sudipto Roy, Rutuja Patil, Aditi Apte.

**Funding acquisition:** Samira Aboubaker, Shamim Ahmad Qazi, Rajiv Bahl.

**Investigation:** Sudipto Roy, Kavita Thibe, Arun Dhongade, Bhagawan Pawar, Shamim Ahmad Qazi, Archana Patil.

**Methodology:** Sudipto Roy, Yasir Bin Nisar, Samira Aboubaker, Rajiv Bahl, Sanjay Juvekar, Ashish Bavdekar.

**Project administration:** Sudipto Roy, Kavita Thibe, Arun Dhongade, Bhagawan Pawar, Archana Patil.

**Resources:** Rutuja Patil, Archana Patil, Sanjay Juvekar, Ashish Bavdekar.

**Supervision:** Rutuja Patil, Sanjay Juvekar, Ashish Bavdekar.

**Visualization:** Sudipto Roy, Rutuja Patil, Aditi Apte.

**Writing – original draft:** Sudipto Roy, Rutuja Patil, Aditi Apte.

**Writing – review & editing:** Yasir Bin Nisar, Samira Aboubaker, Shamim Ahmad Qazi, Sanjay Juvekar, Ashish Bavdekar.

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
