## [Decision Letter · Decision Letter 0]

22 May 2020

PONE-D-20-03227 Simplified management of young infants with possible serious bacterial infection (PSBI) when referral is not feasible in tribal areas of Pune district, Maharashtra, India: Implementation research for demonstration of feasibility and scale upDear Dr. Roy,

Thank you for submitting your manuscript to PLOS ONE. After careful consideration, we feel that it has merit but does not fully meet PLOS ONE’s publication criteria as it currently stands. Therefore, we invite you to submit a revised version of the manuscript that addresses the points raised during the review process.

We would appreciate receiving your revised manuscript by May 30 2020 11:59PM. To enhance the reproducibility of your results, we recommend that if applicable you deposit your laboratory protocols in protocols.io, where a protocol can be assigned its own identifier (DOI) such that it can be cited independently in the future. For instructions see: http://journals.plos.org/plosone/s/submission-guidelines#loc-laboratory-protocols

We look forward to receiving your revised manuscript.

Kind regards,

Khin Thet Wai, MBBS, MPH, MA (Population & Family Planning Resear

Academic Editor

PLOS ONE

Additional Editor Comments (if provided):

This is the important study contributing towards the reduction in neonatal mortality in a resource-limited scenario. Authors have identified the service delivery outcome of the simplified management of PSBI among young infants as well as the implementation outcomes that is fidelity and feasibility.

In general, English language correction is deemed necessary.

In addition, authors should fulfil the following requirements to further strengthen the scientific rigour in reporting the implementation research.

1. To include key reference published in WHO Bulletin

Hales S, Lesher-Trevino A, Ford N et. al.

Reporting guidelines for implementation and operational research.

Bulletin of the World Health Organization 2016;94:58-64. doi: http://dx.doi.org/10.2471/BLT.15.167585

2. The authors needed to clarify the existing knowledge gaps for implementation barriers towards the service delivery guidelines in the Introduction section. What was the nature and severity of the problem?

3. P. 6 LINES 119-120: This is a multisite and multi-country study. Please mention other countries supported by WHO to participate in this study to avoid competing interest in publication ethics.

4. To submit tools used for the baseline survey as additional files.

5. To add one Table to illustrate the results of the baseline survey clearly.

2. Please address the following:

- Please ensure you have thoroughly discussed any potential limitations of this study within the Discussion section.

- Please include additional information regarding the survey used in the study and ensure that you have provided sufficient details that others could replicate the analyses. For instance, if you developed a survey as part of this study and it is not under a copyright more restrictive than CC-BY, please include a copy, in both the original language and English, as Supporting Information.

4. Please upload a copy of Supporting Information Table 1-2 which you refer to in your text on page 7 and 14.

5. We note that Figure 2 in your submission contain map images which may be copyrighted. All PLOS content is published under the Creative Commons Attribution License (CC BY 4.0), which means that the manuscript, images, and Supporting Information files will be freely available online, and any third party is permitted to access, download, copy, distribute, and use these materials in any way, even commercially, with proper attribution. For these reasons, we cannot publish previously copyrighted maps or satellite images created using proprietary data, such as Google software (Google Maps, Street View, and Earth). For more information, see our copyright guidelines: http://journals.plos.org/plosone/s/licenses-and-copyright.

1.    You may seek permission from the original copyright holder of Figure 2 to publish the content specifically under the CC BY 4.0 license. 

Reviewers' comments:

Reviewer's Responses to Questions

**Comments to the Author**

1. Is the manuscript technically sound, and do the data support the conclusions?

Reviewer #1: Partly

Reviewer #2: Yes

2. Has the statistical analysis been performed appropriately and rigorously? 

Reviewer #1: Yes

Reviewer #2: Yes

3. Have the authors made all data underlying the findings in their manuscript fully available?

Reviewer #1: Yes

Reviewer #2: No

4. Is the manuscript presented in an intelligible fashion and written in standard English?

Reviewer #1: Yes

Reviewer #2: No

5. Review Comments to the Author

Reviewer #1: The authors should be congratulated on the execution of an important implementation study for LMICs. Of note, this implementation programme is noteworthy for its attention to sustainability, and the write-up for its detailed description of the programme that fosters reproducibility.

This study has many important strengths that make this manuscript a valuable addition to the literature on the topic. The authors describe the successful implementation of identification and management of PSBI in young infants in concert with the public health system. The clear roles of the research team versus the public health system support the sustainability of this effort.

Despite these strengths, the manuscript requires major revision with respect to its conclusions. The programme demonstrates successful outpatient management of young infants with FB from 7-59 days. This population typically does not warrant referral per WHO criteria. However, the authors overstate what can be drawn from this programme regarding outpatient management of PSBI when referral is not feasible. Only 9 cases in this cohort fell into this category;

As such, attention should be paid to revising the following:

1) Full Title should be revised; the short title is more appropriate, as the bulk of the evaluation is around feasibility of identification and management of PSBI when referral is possible, or when referral is not indicated

2) Abstract should be revised based on the revisions made to the rest of the manuscript

3) The extremely low rate of pre-referral doses of gentamicin is concerning. This does not bode well for MOs/ANMs being confident in administering gentamicin for young infants when referral is not feasible. Among the 9 infants treated as outpatient, only 2/3 received all gent injections. More attention should be given to this problem of lack of confidence in the discussion.

4) the facilitators and challenges faced in the results section could be strengthened by a systematic look at facilitators and barriers using a validated implementation science framework such as the Consolidated Framework for Implementation Research. This would reduce bias in the results presented.

5) The discussion needs to be re-written.

a. paragraph 2 of the discussion: this conclusions should be placed in the context of prior literature on the topic. Do prior studies with larger sample size already demonstrate FB in 7-59 day young infants could be effectively treated on an outpatient basis? If their sample sizes were smaller, cite them. If they were larger, reframe the importance as demonstrating it is FEASIBLE to enact this recommendation with the kind of programme you implemented.

b. line 472---these numbers are too small to make conclusions about helping to reduce neonatal and infant mortality. soften this statement.

c. paragraph starting with line 475--coverage with this programme is a HUGE strength. more attention should be paid to this aspect. what was key in this programme with respect to improving how providers identify infection?

d. paragraph starting with line 479--also important to emphasize that the high acceptance of referral could limit generalizability of this study to settings where such high referral is not possible

e. line 512: is it fair to say heistancy was 'minimized?' the data would suggest that they remained hesitant given the very low numbers or pre-referral gent administered

f. paragraph on line 529--this reference is randomly inserted. please add text linking why this is relevant to the current study

6) Conclusion is overstated. you demonstrated id and simplified management of PSBI and pneumonia in young infants can be feasibly implemented in a population where referral is nearly always possible. re-word.

Additional revisions recommended:

1) Figure 1: both the RST and IT 'provided training' under activities; clarify in the figure how this was different ie what trianing and to whom?; the description about the TSU is vague--ie what recommendations were being provided?

2) Line 176: Is there a zero missing from this number? "1,40,000"

3) In general, all figures on the PDF I reviewed were blurry. Legibility should be improved.

4) Figure 3: This seems more like a flow chart to guide clinical identificaiton and management of young infants with PSBI rather than a flow chart of 'study activities.' Suggest re-labeling.

5) Lines 311 and 313 seem contradictory. what percentage of ASHAs were aware of danger signs in young infants?

6) line 330--did the public health system or RST provide weighing scales, thermometers etc.?

7) line 343: do you mean 545 young infants were identified as potentially sick during the implementation phase? please clarify

8) Table 4: the oral amox doses section of the table is confusing. Does received all 14 days of treatment refer to 7 days of gent and 7 days of amox? many of these rows do not apply to the FB 7-59day set of patients who only get 7 days of amox. Also please clarify what is meant by the comment associated with the * in this table.

8) line 367: this sentence seems repetitive of the prior one in which it states on infant had CI among those treated on an outpatient basis

9) #3 in the table on challenges--it is unclear until you get to the discussion why is is problematic that many parents preferred private providers; add some explanation to clarify in the results section

10) #6 in the table on challenges--the role of the RST here seems like one the public health system should have taken on? this may warrant comment in the discussion

11) Figure 4: label the y-axis

Reviewer #2: Description of the study area has to be exhaustive. Number of staffs, their level of training and the referral linkages has to be clearly outlined. Coverage calculation has to be justified as per the comment given.

Supplemental tables are not attached or could not be downloaded. The whole manuscript needs to be checked according to the PLOS guideline. Font size, spacing,title and tables has to be revised.

6. PLOS authors have the option to publish the peer review history of their article (what does this mean?). If published, this will include your full peer review and any attached files.

Reviewer #1: Yes: Jacquelyn K Patterson

Reviewer #2: No

---

## [Author Response · Author response to Decision Letter 0]

15 Jun 2020

We, the authors, thank the reviewers and the journal editorial team for their valuable comments and suggestions, which have helped to improve the quality of the manuscript. We have responded to all the comments and have incorporated the suggestions. Our responses to the comments are marked in blue in the attached document on responses to reviewers. We have also replied to the comments in the revised manuscript with track changes. 

Reviewer#3 (Abadi Leul)

Introduction 

• Lines 68 – 69: Reference number five says 92 % of all neonatal infections (not illnesses) are caused by bacterial infection. Neonatal infection is responsible for 46% of neonatal illnesses. Check and correct please. 

We have corrected this as: bacterial infections accounted for 92% of the known causes of death due to possible serious bacterial infections. For better clarity, we have now also included mean incidence of bacterial infection (13.2% per 1000 livebirths). 

• Line 83: Here please either mention all the regimens used in the AFRINEST and SATT studies or remove either procaine penicillin or Amoxicillin. Mention also Pneumonia in 7 – 59 days can be managed effectively without the need for referral as you have implemented this approach as well.

We have modified the text appropriately by removing the mention of penicillin and amoxicillin. The revised text is as follows: 

“Subsequently, a series of several large community based, multi-centre randomized trials in African and Asian populations (known as AFRINEST [African neonatal sepsis trial] and SATT[Simplified Antibiotic Therapy for Sepsis in Young Infants]) demonstrated the effectiveness of simplified antibiotic regimens containing injectable gentamicin, procaine penicillin and amoxicillin in out-patient settings among neonates and young infants up to two months of age with signs of PSBI where referral was not feasible.”

• Lines 86 – 87: Please check your reference. Reference number 17 and 18 is about home based new born care but could not get the section which describes about PSBI.

• Thank you for pointing this out. We have replaced the references with the following references. 

Reference no. 18. Government of India. Operational Guidelines for use of Gentamicin by ANMs for management of sepsis in young infants under specific situations. Ministry of Health and Family Welfare, Government of India, Feb 2014. Available at http://tripuranrhm.gov.in/Guidlines/2606201401.pdf accessed 26th May 2020. 

Reference no. 19. Government of India. Addendum to Operational Guidelines on Use of Gentamicin by ANMs for Management of Sepsis in Young Infants under Specific Situations. Available at https://nhm.gov.in/New_Updates_2018/Om_and_orders/rmncha/child_health/Addendum.pdf accessed 26th May 2020. 

Methods

• Lines 113 – 114: The outcome is not clear. What do you mean when you state “All or at least 80% of identified sick young infants will receive treatment.” This outcome is not clear. It means for me whatever the rate of identification you will provide at least 80% treatment. One of the challenges in management of neonatal infections is most of them are not being identified. It is not problem of provision of treatment if identified. Of course referral acceptance is an obstacle to get treatment. I would suggest to re write the two out comes as objective

1. To identify at least __ % of all expected PSBI cases

2. To provide adequate treatment to at least ___% of those identified.

The same lines better to avoid all or at least are enough

We agree that an estimate of the coverage of identification is an important indicator. We reported these objectives as such because they were listed as the objectives in our study protocol. However, we have now modified the text as follows: 

“At least 80% of identified sick young infants will receive treatment.

At least 80% of the identified sick young infants will receive adequate treatment (Adequate treatment was considered as antibiotic doses for the first two days followed by at least 70% of doses over the next five days.)

We have calculated coverage and discussed about it in the sections on results and discussion.

• Line115: I see no difference in defining adequate treatment as taking full doses of amoxicillin (10 doses) and five days of injectable gentamicin. In case there is a difference elaborate.

We have explained this in the section on implementation of simplified management of sick young infants below and in the objectives kept it as ‘all doses of recommended treatment’. 

Design

• Lines 124 – 125: Which WHO guideline is this. Put a reference. 

• There are also two options that I know in the WHO guideline. It seems that you choose to use the seven days regimen that comprises 7 days gentamicin injections and twice oral amoxicillin per day for 7 days. Provide justification to use this regimen.

Please see below the response to both these comments: 

The WHO reference we used was:

Reference no. 22. Integrated Management of Childhood Illness: management of the sick young infant aged up to 2 months. IMCI chart booklet. Geneva: World Health Organization; 2019. License: CC BY-NC-SA 3.0 IGO. 

This regimen was chosen after discussions with the Government of India, which recommends the seven-day antibiotic regimen for PSBI. 

References: 

Reference no. 18. Operational Guidelines for use of Gentamicin by ANMs for management of sepsis in young infants under specific situations. Ministry of Health and Family Welfare, Government of India, Feb 2014. Available at http://tripuranrhm.gov.in/Guidlines/2606201401.pdf accessed 26th May 2020. Reference no. 19. Addendum to Operational Guidelines on Use of Gentamicin by ANMs for Management of Sepsis in Young Infants under Specific Situations. Available at https://nhm.gov.in/New_Updates_2018/Om_and_orders/rmncha/child_health/Addendum.pdf accessed 26th May 2020

We have also added the following in the section on ‘Trainings and Orientation’: The algorithm for identification and management of PSBI was prepared from adopting updates from the WHO PSBI guidelines guideline on managing PSBI in young infants when referral is not feasible to the WHO young infant IMCI chart booklet, 2019. Although the WHO young infant IMCI chart booklet was printed in 2019, the draft was used to adapt the Indian guideline to manage PSBI when referral was not feasible.

Policy dialogue for implementing adapted guidelines

• Line148: Could not see supplemental table 1.Is it in a different attachment than the figures.

We seem to have missed to include this supplementary table initially, have added this along with the re-submission.

• Line 158 – 159: What is the difference between RST and TSU? Don’t they have the same task? It is not clear why you divided them into two. I feel the RST is in the TSU. If that is the case IT and TSU may be enough to make things clear.

The TSU was an oversight body with senior independent experts and study investigators. It was not involved in the conduct of the study, it only met four times during the study to provide guidance and suggestions. The RST was involved in day-to-day study activities, including supporting the implementation team (IT), regular supervision and analysis. Thus, the TSU was completely independent of the RST. We have now clarified this in the text. 

Study area 

• Better to re write to keep the flow. Also give a summarized picture of the study area. Total population covered. Expected live births in the catchment area in 20 months

.Available health facilities from the lowest level to the highest level in the district. How are they staffed under normal circumstances based on the government policy e.g., What are the staffs available at the sub centers PHC, rural hospitals and district hospital? What can they do? Distance between each level of care and catchment population for each tier level.

Thank you for this suggestion. We have added details of the study area in a panel in figure 3 as well as a table (supplementary table 2) that gives a summary of the study area. 

• Line 177: check the number.

Yes, we have confirmed this number. As this is a tribal area, it is relatively less densely populated

Baseline survey and mapping of referral system

• General the following should have been included in the base line assessment 

The existing services in the PHC and sub center? Activities in relation to PSBI cases? Whether they are identifying these cases and what do they do.

You should have also assessed the MOs the same way as ANM/ASHA

We have included these details in a panel in figure number 3

• Line 185: what are ANMs? Don’t start with abbreviation in the first time.

We have explained above in a panel in figure number 3

• Line 86 – 87: what are ASHAs? You didn’t describe them before?

We have explained above in a panel in figure number 3

• Line 194: IT team activities. I don’t see the importance of this title. You can continue with the training and orientation you are describing about training of the master trainers from the TSU.

We have removed the title “IT team activities”.

• Line 200: Reference 21 is the 2019 WHO guideline. The training was given in 2017. Please recheck this.

We have explained this as: The algorithm for identification and management of PSBI was prepared from adopting updates from the WHO PSBI guideline on managing possible serious bacterial infection in young infants when referral is not feasible (17, 22) to the WHO young infant IMCI chart booklet, 2019 (23). Although the WHO young infant IMCI chart booklet was printed in 2019, the draft was used to adapt the Indian guideline to manage PSBI when referral was not feasible. 

• Line 206: Fig. 3 Needs rearrangement. Based on the WHO guideline PSBI includes FB in young infants 7 – 59 days as well. Please check and correct.

We have changed the figure accordingly [now figure 4].

Identification of sick young infants 

• Line 217: What is this higher level? You have Rural and district Hospitals. Are these the referral sites or you have a tertiary level. If different one please provides its distance and the neonatal services in this level somewhere.

We have included this in a panel in figure number 3: Civil hospital, Pune: This is a district-level tertiary care hospital with neonatal intensive care facilities and is located in Pune district at an average distance of 80-140 kilometres from the study area. This was the primary referral centre in the study.

Referral to tertiary health care facilities 

• Line 225: It is not clear why you were bypassing the rural and district hospitals. Please explain why this happened. What is the purpose of these hospitals if they don’t serve as referral sites for PSBI cases?

We have added in the text: The rural hospitals and sub-district hospitals only manage uncomplicated infection cases in infants, and provide immediate post-natal care but and do not have neonatal intensive care facilities to and do not admit or manage serious illness. PSBI cases. Hence these were not considered as referral facilities for PSBI cases. They have been described here to give a complete picture of the health infrastructure.

• Line 226: Was this the pre referral regimen for India. Is it different from the WHO recommendation which was giving gentamicin and oral amoxicillin?

As per the Indian study guidelines, only intramuscular Gentamicin was given as pre-referral antibiotic dose.

Treatment when referral is not feasible or referral is not recommended

• Lines 234 – 236: Are you sure you gave oral amoxicillin and gentamicin injection for CI. Is this in your guideline? Please separate CSI from CI. I feel it is by mistake. You may give ampicillin injection instead of amoxicillin?

This was an error made initially; we have now mentioned the appropriate treatment for CI. Infants with CI were started on a course of intramuscular ampicillin and intramuscular gentamicin daily for seven days. 

• Line 237: Why is the MOs work only consultation? These are the most vulnerable babies and they should get the maximum care by the most senior person in the PHC. 

We agree and in our study the MOs had overall responsibility for identification of PSBI, choosing the appropriate treatment and administering the treatment. However, in many cases, once the first dose of antibiotics was administered, the infants would follow up with ANMs, who were fully trained to continue to administer the antibiotics. 

We have now modified the text to “All doses of injection gentamicin and first daily dose of oral amoxicillin were administered by the MO or ANM in consultation with the MO at the PHC”. 

Monitoring and follow up

• Line 246 – 248: Who was doing the follow up and where? Why only on day 4 for FB pneumonia. Was there any follow up after they finish their treatment to know the outcome?

Thanks for pointing out this error. We have now corrected as: all FB cases were followed up on the 4th and 8th days of starting treatment.

Activities of the KEMHRC RST

• Line 263 – 264: Please state how the consultation was made? And whether it was random consultation or regular for every case? If regular how often?

Consultations were made for approximately 10% of all PSBI cases, not for all cases. The consultations were always initiated by the MOs/ANMs. There was no pre-decided pattern for consultations. We have updated this in the text. 

Communication 

• Line 278 – 280: Please mention here how frequent was this done? How many cases were assessed using this application? How accurate were they in classifying and identifying treatment? Better if you have data as this is one of the best ways of remote consultations? 

Whatsapp technology was used for consultation in 5-6 cases of PSBI. However, this was done after PSBI classification was done by MOs/ANMs and only for confirmation. We did not asses the validity of this technique in the study, hence we cannot comment on its scientific accuracy. We believe that this is a technology for remote consultations in our set-up. We agree that this may need separate validation studies, but it was out of scope of this project. 

• Line 279 – 280: How frequent was this meeting held. How did it help in bringing changes?

We have added this in the text: These meetings were held once every month for the initial five-six months as most of the challenges were identified during these initial months into the study and thereafter, once every three months. These helped in devising solutions to improve case identification, ensuring uninterrupted supplies of medicines and consumables and ensuring referral mechanisms could function appropriately. 

Results

This section has important findings but needs major revision. Coverage calculation seems to be from the actual cases identified. As well all know one of the major challenges in the management of PSBI in resource limited countries like India is identification of these newborns. Hence calculation of coverage is very important. The coverage depicted here shows only coverage for identified. The key finding and informative for policy change will be identification from the expected live births. Hence you need to indicate in your study area (please look comment in the section study area description). Look also comments in each section.

• Line 294 – 296: Need to rephrase it as

The public health system has successful managed 90% of the PSBI cases identified during the study period independently while 10% required consultation with the RST implying the study was implemented as planned.

 We agree with this statement but found it relevant for inclusion in the discussion section and have added this statement in the first paragraph in discussion section.

Baseline Survey: 

• Based on your findings please show the referral system (How these different health facilities are linked each other. Which once are the referral sites for PSBI cases? Why the secondary level of health care was were bypassed. How are the private institutions staffed? Also describe their exact number. How few are few?

We have depicted the referral system in figure 3. The rural hospitals and sub-district hospitals do not have neonatal intensive care facilities and do not admit or manage PSBI cases. We have now mentioned in the methods section. Hence these were not considered as referral facilities for PSBI cases. These hospitals manage only uncomplicated infection cases in infants and immediate post-natal care. 

There were approximately 8-10 private health facilities, within the study area, with 2-3 some having intensive care units. We have updated this in the text. We did not perform a health facility survey for private health facilities as part of this study, hence we do not have the exact numbers. These private health facilities varied in terms of their capabilities, with in-patient capacity ranging from 5-20 beds and staff consisting of a mix of doctors, nurses and support staff. 

• Supplemental table 2. I could not access it. Reload it or indicate where it is. At least not with the attached supplements.

We seem to have missed to include this supplementary table initially, have added this along with the re-submission

• Line 312 – 321: This paragraph needs to be re written clearly. Some of the data are not uniform example you said “….about 80%-85% ASHAs were aware about danger signs in young infants and counselling for newborn care. Then ….. Only 55-60% ASHAs was aware of danger signs in young infants” 

Thanks for pointing out the error. We have corrected it as: 80%-85% ASHAs were aware about overall care of young infants and counselling for newborn care However, Only 55-60% ASHAs were aware of danger signs in young infants.

• The ANMs have never identified PSBI cases before. Does this mean no NB was coming at all or how what were they doing when a new born comes to them? Clarify in order to understand.

Have added this: Before initiation of this project, sick young infants approaching the PHCs were routinely referred to higher-level health facilities by MOs. 

• Line 326: Mention the strength of Gentamicin available at that time?

Have added: vials in a dose of 80mg/2ml.

Birth surveillance and postnatal visits

• Do we know how much of the expected live births were identified? This is the main reason for compromising identification of PSBI cases as seen in other studies. According to the World Bank 2017 report the estimated live births in India was 18.08/1000population. Hence your area should expect 139942*0.01808/year which is equivalent to 4217 LB. You have identified only 2001. I assume that many might have delivered in the rural and district hospitals as they are relatively accessible. You may be able to estimate the proportion of the mothers who had follow in these hospitals and proportion missed taking the local circumstances. Taking this into consideration you can estimate the coverage otherwise it does not give sense calculating from the identified. Infact it should have been 100% if you calculate from the identified. I feel this has also affected your coverage of PSBI. I recommend you to give justifications for this.

During the study period of August 2017 to March 2019, the study recorded totally 3071 live births in all institutions. Of these 3071 births, 2001 were in public health facilities and were followed up for post-natal home-based care by ASHAs. Based on this number of 3071 live births, we could follow 2001 (65%) births. We have updated this in the text. 

• Table 1: Title Follow the guideline given by PLOS.

We have changed the title

• Either you can omit this table and describe or better if you add coverage from expected. 

We have added a column and moved this table to the supplementary tables document.

Identification and referral of sick young infants 

• This section requires major revision

Your result mainly focuses on treatment not on identification. If I am correct WHO needed this implementation research mainly how to identify young infants with PSBI and then provision of adequate treatment? Your result in terms of provision of treatment is very nice. As I can see from your main aim of the research as well is focusing only on treatment. I was expecting coverage as well. I don’t think the number of PSBI cases expected was only 200 in 20 months in this district. If you correct the expected live births then you will get the answer. I am thinking many PSBI cases have gone directly to the private clinics and government hospitals. Is it possible to get this data or at least put estimation? Unless we have that data it will be difficult to accept 87.5% coverage of identification in this catchment.

The other issue is that in your description it seems that FB pneumonia in young infant 7 – 59 days is not considered as PSBI. Hence this part needs to be re written. 

Have reworded this: A total of 545 sick young infants were assessed during the study implementation phase, of which 175 90 had signs of PSBI (34 CI, 46 CSI, 10 FB in 0-6 days old infants and 85 infants aged 7-59 days with FB). Assuming a 10% incidence (5, 13, 14, 15, 16) of PSBI among all live births, with 3071 live births recorded, the expected number of young infants with any sign of PSBI or including FB in 7-59 days old would be 307.We identified and treated 175 young infants with PSBI, indicating actual coverage of 57%.

We did not collect data from private clinics and government hospitals as this was outside the study protocol. We agree that some infants would have been identified at these hospitals and the actual coverage may need a correction factor. However, we have not attempted to estimate this factor in this study. The purpose of this study was to demonstrate the feasibility of implementing the PSBI management intervention in the public health system. Coverage by this public health system, though an important indicator, may not be very high in places where private health facilities are available.

• Lines 351 – 354. You have young infants sent to the sub centers. Can you separate those sent to the sub centers and to the PHC and then describe how they were managed and/or referred. The same is true as to who evaluated them MOs/ ANMs. Do we know how many were evaluated by MOs and ANMs separately? Do we know the reason why all were not seen by the MOs if these are the most senior people you have in the PHCs. 

This is an important piece of information and we appreciate the questions on this. However, while we have data on pathways to treatment, we have deleted the pathways here since that in itself is a separate analysis. It includes at least five different pathways to treatment seen in the study and would greatly increase the content of this manuscript. For example, each pathway would have person and place of identification, followed by person and place of first assessment, and sometimes, second assessment, person and place of starting treatment and outcomes. This comment, though, does give us an idea to conduct this analysis and write a separate manuscript; we will follow-up on this and thank the reviewer for this suggestion.

We have reworded this section to only specify identification.

• Table 2. Follow guideline given by PLOS and correct the title of the table accordingly.

We have modified accordingly. (Now table 1) 

• Line 366 – 368: Mention whether all have received pre referral treatment? If yes what pre referral treatment was given?

Have added to the text: Pre-referral doses of gentamicin were given in only 7.1% (6 out of 84) PSBI before referral to higher centres.

• Line 378: Correct as 175 not 185.

Corrected to 175.

• Line 380: WHO doesn’t recommend very low birth weight to be managed as PSBI at community level and it was one of the exclusion criteria in the AFRINEST and SATT trials. Does your implementation include VLBW? 

VLBW infants were supposed to be referred and not managed at PHC as per protocol.

• Line 382: Which one is your tertiary referral level? Is it the district hospital? If that is considered the tertiary referral level please describe some where the referral site is the district hospital.

We have added sentences in the methods section of the manuscript explaining that the Civil hospital was the referral hospital. The rural hospitals and sub-district hospitals do not have neonatal intensive care facilities and do not admit or manage PSBI cases. Hence these were not considered as referral facilities for PSBI cases. These hospitals manage only uncomplicated infection cases in infants and immediate post-natal care. Have also mentioned in the newly added section on health infrastructure that the Civil Hospital is a district-level tertiary care hospital with neonatal intensive care facilities and is located in Pune district at an average distance of 80-140 kilometres from the study area. 

• Line 383 – 384: Just wondering how he received treatment at the PHC. Was he able to take the PO medication> state the medication that he took. Give the reason why the mother refused treatment on the second day. How comfortable was the HCP to treat such babies.

One received two doses of oral amoxicillin and no injection gentamicin and the other one received three days treatment with injection gentamicin and oral amoxicillin. The MO/ANM were prepared to administer complete treatment for 7 days, however both the parents refused treatment without giving any specific reasons.

• Table 3. The same follow PLOS guideline

We have modified the table accordingly. (Now table 2)

• Better if you remove the column on FB pneumonia. It is misleading. It says 22 accepted referral which is 26%. These are not offered referral rather did not accept the treatment at PHC. 26% is big. Try to explain why they have declined to be treated at the PHC rather.

We have removed data for FB in 7-59 days and coloured the column grey. Have discussed this in the discussion section.

• One young infant with FB pneumonia came to the PHC and refused both treatment and referral. It is interesting to know why he came and why he refused every offer. It is unusual; finding and better to explain if we know the reason.

The reason was that the parents had planned to immediately migrate out of the study area and did not want to start treatment from the PHC.

Treatments given to sick young infants when referral was not recommended or feasible 

• Please separate CSI and CI who refused referral and those with FB who were treated without offering referral to make it more informative.

We have presented results separately for CI, CSI, FB in 0-6 days when referral for not feasible and for FB in 7-59 days where referral is not recommended.

• Line 401 – 402: Please explain why pre referral treatment was given only to 6/81 young infants. Why was it not corrected during the process of implementation?

Have moved this to section on Referral to government or private tertiary care and management of sick young infant at PHC/SC in the manuscript. This has been discussed as a challenge faced in the study.

• Line 402 – 404: This is contradictory to what you have reported in those who did not accept referral. You stated that they have treated VLBW infants. If the ANMs/MOs were not trained to give Im gentamicine, who gave treatment to the one baby who refused referral for one day. Check this with line 382 – 384. This needs explanation or correction. 

MOs/ANMs were trained to give Gentamicin and as seen in table 4, have given gentamicin to 7 PSBI infants who refused referral. However, this was a specific incidence at one PHC and have clarified that MOs/ANMs of that particular PHC refused to treat such very low birth weight babies

• Line 403: Put the exact weight than saying <1.5 kg

Have done accordingly.

• Line 405: Does that mean he was cured without any intervention. You can leave this as there is nothing to learn from this. 

Have added: This infant had hypothermia and movements were lesser than normal. Breastfeeding was initiated and continued adequately.

• Table 4. Follow PLOS guideline.

Done accordingly (Now table 3)

• Fb only Pneumonia in 7 – 59 days: Is it 63 or 62? Check and correct.

It is 62, have corrected this.

Study Progress and facilitators and challenges for implementation 

• General comment on this issue is related to the results. Unless we get the number of young infants going directly to the rural Hospitals and district Hospitals which I feel is the case. It is very difficult to generalize that your case identification has improved. Still this section needs revision if the result you put is the same. It is expected that there are at least equal number of babies that you failed to identify considering the PSBI cases that you expect in the district. But it may be true that most of them have gone directly to the nearby hospitals or went out of the catchment. I recommend to explain this.

We appreciate this suggestion. We did not collect data from these hospitals as this was outside the study protocol. We agree that some infants would have been identified at these hospitals and the actual coverage may need a correction factor. We have added this in the discussion section.

• Fig 4. Needs rearrangement. PSBI includes all. Hence better to change the legend as Total PSBI, CSI/CI and the FB pneumonia in 7 – 59 days. Then the increment is on FB pneumonia while the CSI/CI is declining. How do we explain this?

We have done accordingly.

• Line 441 – 444: In your IR it seems that the rural and district hospitals were by passed to a tertiary level. If no PSBI case was being seen before the IR then where were these babies seen? If they were seen in the rural and district Hospitals, this is going to be a shift from a relatively equipped health facility to a lower and less equipped facility. The activities of these hospitals were not explained. Do you have this data? 

We have added details of these hospitals and why these were not referral centres in the methods section

• Line 445 – 446: What does 53.3% indicate? The eligible ones to be referred were 90 PSBI cases and of which 90% accepted referral (line 365). Check this and keep consistency in descriptions.

We have modified as follows: A robust and effective referral system ensured that a majority [81 out of 90 (90%)] of young infants with PSBI, who were eligible for referral to government tertiary care health centres, actually accepted referral to such centres

Challenges faced

• Table 5: As in other tables.

Done accordingly. (Now table 4)

• Row number 3: Why was it a challenge if they want to go to private centers as far as they provide appropriate treatment? Do we know their capacity to consider this as a challenge? You did not show their limitations in your description of study areas. In fact they should have been included in this IR as many babies might have gone there.

While we did not evaluate private health facilities in the study, we cannot say anything about quality of care at private facilities or appropriateness of their treatment. Further, from studies on antibiotic prescriptions, it has been seen that very few private facilities treat infections in children with simplified antibiotics such as gentamicin and amoxicillin, with most preferring cephalosporins (28, 29). In addition, we are not sure if these were affordable for the study population. This aspect needs further investigation and we agree that the private health sector, where available, can play a valuable role in managing sick young infants. We have discussed this aspect in the discussion section. 

• Row 4: Do you think you have solved this challenge? Only 6 babies received prereferral treatment out of 81 not 11 by the way. (Correct this). How come you are satisfied to say you have solved this challenge? It seems it has persisted throughout the implementation processes. Give explanation

Have corrected the number. We have clarified that this was not satisfactorily resolved in the study in the discussion section. 

• Row 6: During which phase was this?

This was seen in the first three months of the study.

• Row 7: It is confusing. The nurses where giving gentamicin for VLBW who did not accept referral but not for pre referral. Rationale? Most countries including WHO guideline doesn’t include VLBW in these classifications. You did not mention about the difficulty of giving amoxicillin for these babies whom I feel is more difficult.

Have clarified this: MO/ANMs were trained in administering injections even to VLBW babies as part of routine immunizations; they were not confident of administering gentamicin for fear of adverse events. The MO/ANMs did not report any difficulty in administering amoxicillin to VLBW babies.

Discussion

• General comment 

The discussion part is well discussed but the problem lies on calculating coverage. Unless otherwise explained the coverage of identification of PSBI in this district is not satisfactory. May be the private centers and the three hospitals are taking major share. Here either include data captured from these sites or assume based on reasonable reasoning that certain amount has gone to these sites. Please re calculate the coverage for identification of PSBI. See comments in result section.

Have responded to these comments partly in the results section.. 

Have also added the following in the discussion section: We had reasonable treatment coverage of a little less than 60% in our study. Similar implementation research studies on PSBI in young infants from African and Asian countries reported treatment coverages of 16.3% in Sylhet and Chittagong in Bangladesh (23), 50% in Ethiopia (24), 42% in Kushtia district, Bangladesh (25), 63.8% in Malawi (26) and 95% in Nigeria (27). Thus, except for the Nigeria study, our coverage proportions are comparable with other reported coverages. However, our coverage could have been higher; a major factor in our study area was the preference of private health facilities which might explain where the remaining estimated 40% cases of PSBI would have been managed. Another factor could be that some PSBI cases visited the sub-district and district hospitals directly. However, we did not collect data from these hospitals as this was outside the study protocol. We agree that some infants would have been identified at these hospitals and the actual coverage may need a correction factor. 

• Line 492 – 495: This part is good> If you had this information that even those who came to the PHC were preferring the private facilities the you should have collected data from those who went direct to the health facility. This is important as the main objective off WHO is identifying all young infants with PSBI and make sure that they receive adequate treatment including simplified antibiotics no just getting treatment at the PHC.

MO/ANMs were able to ascertain preference for private facilities only after these cases came to SC/PHC and were diagnosed with PSBI. There was no scope for separate data collection from those who directly went to private facilities. Primary reasons given for private preference were relatively more confidence in and relative ease of negotiating private health-care processes. In fact, this finding from the study will help to inform policy regarding PSBI management when referral is not feasible to a larger public sector hospital, wherein greater public-private partnership in health can be a recommendation to reduce infant mortality. However, from studies on antibiotic prescriptions, it has been seen that very few private facilities treat infections in children with simplified antibiotics such as gentamicin and amoxicillin, with most preferring cephalosporins (28,29). In addition, affordability of such facilities remains a challenge, especially for the tribal communities as in the study area. Thus, a balanced view about the appropriateness, affordability as well as accessibility of private health facilities needs to be considered.

• Line 503 – 504: Do we know the reasons why we did not achieve our expected output which was to provide at least 80% quality treatment? Discuss some explanations

Have reworded and added in the text: The quality of treatment received at PHC level was appropriate for around 65% of the CSI, CI and FB in 0-7 days PSBI cases in those who accepted the treatment. While this was based on only 9 cases for which referral was not possible, we admit that even for such a small number, the quality should have been higher.

• Line 507 - 509 – was this problem continued till the end or remained a problem. Explain this. If this remains to be a problem then how do you see the ownership of this programme?

This problem did not continue as the confidence of primary healthcare workers mainly ASHAs and ANMS increased with time for identification and treatment of the sick infants. Relevant statement is added to the discussion.

• Line 538 – 541: Another way of looking effectiveness of such programmers is the change in the identification and treatment coverage of young infants with PSBI in the whole district compared to the previous years other than looking in to the increment at the PHC level only. How do you see your involvement in the community training on newborn danger signs by the ASHAs? Did it bring improvement in care seeking and more and more kids are going to the other centers as well or are you bringing those who used to go to the other centers in to the PHC. If the later is the case this is not what is expected. In case you have data try to look into this as well.

We agree that this information would add value to the interpretation of our study results. We did attempt to obtain this data from other facilities in Pune district during the study, however we could not obtain any relevant data. Similarly, we did not measure care-seeking practices during this study; however, we, KEM Hospital Research Centre, are currently conducting a separate care-seeking study in under-five children in the same tribal areas and will get an estimate of whether the PSBI study led to any change in care-seeking preferences and practices.

Reviewer #1:

 The authors should be congratulated on the execution of an important implementation study for LMICs. Of note, this implementation programme is noteworthy for its attention to sustainability, and the write-up for its detailed description of the programme that fosters reproducibility.

This study has many important strengths that make this manuscript a valuable addition to the literature on the topic. The authors describe the successful implementation of identification and management of PSBI in young infants in concert with the public health system. The clear roles of the research team versus the public health system support the sustainability of this effort.

Thank you for your appreciation. 

Despite these strengths, the manuscript requires major revision with respect to its conclusions. The programme demonstrates successful outpatient management of young infants with FB from 7-59 days. This population typically does not warrant referral per WHO criteria. However, the authors overstate what can be drawn from this programme regarding outpatient management of PSBI when referral is not feasible. Only 9 cases in this cohort fell into this category;

We have modified our conclusion accordingly. Also, in the discussion, we have included few numbers pf PSBI cases being managed at PHCs as one of the limitations. 

As such, attention should be paid to revising the following:

1) Full Title should be revised; the short title is more appropriate, as the bulk of the evaluation is around feasibility of identification and management of PSBI when referral is possible, or when referral is not indicated

Have changed full title to: Feasibility of implementation of simplified management of young infants with possible serious bacterial infection (PSBI) when referral is not feasible in tribal areas of Pune district, Maharashtra, India

2) Abstract should be revised based on the revisions made to the rest of the manuscript

Have revised the abstract accordingly.

3) The extremely low rate of pre-referral doses of gentamicin is concerning. This does not bode well for MOs/ANMs being confident in administering gentamicin for young infants when referral is not feasible. Among the 9 infants treated as outpatient, only 2/3 received all gent injections. More attention should be given to this problem of lack of confidence in the discussion.

Have highlighted this as a limitation and discussed the following in the discussion section: The hesitancy of ANMs to administer injectable gentamicin in young infants due to fear of adverse effects could not be was minimized sufficiently in spite of with refresher training and constant on-field support. This needs further examination and appropriate steps to improve this intervention of administering gentamicin. This was a limitation of this study indicating that there is a need for motivation and confidence-building, including support of senior public health authorities, of MOs, ANMs and ASHAs to administer pre-referral gentamicin.

4) The facilitators and challenges faced in the results section could be strengthened by a systematic look at facilitators and barriers using a validated implementation science framework such as the Consolidated Framework for Implementation Research. This would reduce bias in the results presented.

Thank you, this is a useful suggestion. We also find the CFIR to be an appropriate framework for implementation research and would like to inform that the KEMHRC team is currently conducting an implementation research study using the CFIR to improve outcomes of high-risk pregnancies. However, we believe that using CFIR to determine facilitators and barriers would generate a lot of information that should be presented separately and it needs its own manuscript. We have focused on the overall feasibility and described the primary study findings in the current manuscript which is the first manuscript from the study; we will use your suggestion to present facilitators and barriers in more details in a subsequent manuscript.

5) The discussion needs to be re-written.

a. paragraph 2 of the discussion: this conclusions should be placed in the context of prior literature on the topic. Do prior studies with larger sample size already demonstrate FB in 7-59 day young infants could be effectively treated on an outpatient basis? If their sample sizes were smaller, cite them. If they were larger, reframe the importance as demonstrating it is FEASIBLE to enact this recommendation with the kind of programme you implemented.

Have added: In the Malawi study (already referred to previously in the text), out of 150 young infants with FB, cure rate was achieved in 96% infants (26). The study in Kushtia, Bangladesh reported that among 475 young infants with FB, approximately 80% of caregivers reported that their infant had received oral amoxicillin for five days and 74% for seven days and the treatment failure rate was approximately 5% (25). Similar proportions were reported in the Nigeria study with All 7–59 days old young infants with fast breathing who received treatment at the outpatient improved with oral amoxicillin (27). Thus, our study results had comparative results for outcomes of FB in 7-59 days infants, though, our numbers were lesser. Thus, it is feasible to enact this recommendation within the public health system in India.

b. line 472---these numbers are too small to make conclusions about helping to reduce neonatal and infant mortality. soften this statement.

Have added: However, considering the relatively low number of PSBI cases, we need to interpret this fatality rate with caution when applying to other areas as well as while scaling-up the intervention.

c. paragraph starting with line 475--coverage with this programme is a HUGE strength. more attention should be paid to this aspect. what was key in this programme with respect to improving how providers identify infection?

Have added: Ownership of the program and supervision by senior public health officials were crucial for achieving these coverage proportions. Along with it, support from the KEMHRC RST, especially in the early phase of the study played an important role in this. Have also added comparisons with other similar studies.

d. paragraph starting with line 479--also important to emphasize that the high acceptance of referral could limit generalizability of this study to settings where such high referral is not possible

Have added: The high acceptance of referral could limit generalizability of this study to settings where such high referral is not possible. Have also added comparisons.

e. line 512: is it fair to say heistancy was 'minimized?' the data would suggest that they remained hesitant given the very low numbers or pre-referral gent administered

Have changed this to: However, the hesitancy of ANMs to administer injectable gentamicin in young infants due to fear of adverse effects could not be was minimized sufficiently in spite of with refresher training and constant on-field support. This needs further examination and appropriate steps to improve this intervention of administering gentamicin. 

f. paragraph on line 529--this reference is randomly inserted. please add text linking why this is relevant to the current study

Have moved this reference to the discussion on coverage at the top of the discussion section.

6) Conclusion is overstated. you demonstrated id and simplified management of PSBI and pneumonia in young infants can be feasibly implemented in a population where referral is nearly always possible. re-word.

Have revised as: The study demonstrated that the identification and management of PSBI and their simplified management of PSBI and pneumonia in young infants can be implemented in a safe and effective manner at out-patient facilities in the public health system. when referral is not feasible. Fast breathing in 7-59 days old can be effectively managed with oral antibiotic on an outpatient basis without referral.

Additional revisions recommended:

1) Figure 1: both the RST and IT 'provided training' under activities; clarify in the figure how this was different ie what trianing and to whom?; the description about the TSU is vague--ie what recommendations were being provided?

Have done accordingly.

2) Line 176: Is there a zero missing from this number? "1,40,000"

This figure is correct. As this is a tribal area, it is relatively less densely populated.

3) In general, all figures on the PDF I reviewed were blurry. Legibility should be improved.

Have revised the formatting of the figures.

4) Figure 3: This seems more like a flow chart to guide clinical identificaiton and management of young infants with PSBI rather than a flow chart of 'study activities.' Suggest re-labeling.

Have labeled it now as ‘flowchart of PSBI identification and management’.

5) Lines 311 and 313 seem contradictory. what percentage of ASHAs were aware of danger signs in young infants?

We made an error and have corrected it as: 80%-85% ASHAs were aware about overall care of young infants and counselling for newborn care However, Only 55-60% ASHAs were aware of danger signs in young infants.

6) line 330--did the public health system or RST provide weighing scales, thermometers etc.?

Have added in the sentence: by the RST

7) line 343: do you mean 545 young infants were identified as potentially sick during the implementation phase? please clarify

Have reworded this as: A total of 545 sick young infants with some signs of illness were assessed during the study implementation phase, of which 175 had signs of PSBI (34 CI, 46 CSI, 10 FB in 0-6 days old infants and 85 infants aged 7-59 days with FB.)

8) Table 4: the oral amox doses section of the table is confusing. Does received all 14 days of treatment refer to 7 days of gent and 7 days of amox? many of these rows do not apply to the FB 7-59day set of patients who only get 7 days of amox. Also please clarify what is meant by the comment associated with the * in this table.

14 doses of amoxicillin indicate 7 days of amoxicillin with twice daily dosing. 7 days of gentamicin indicate 7 days of gentamicin with once daily dosing. Have coloured the cells for gentamicin in the column on FB in 7-59 days grey as this is not applicable. We have removed this * sign and comment to avoid confusion; details are written in the text explaining the table. 

8) line 367: this sentence seems repetitive of the prior one in which it states on infant had CI among those treated on an outpatient basis

Have deleted this sentence.

9) #3 in the table on challenges--it is unclear until you get to the discussion why is is problematic that many parents preferred private providers; add some explanation to clarify in the results section

Have added in the table: Primary reasons given for private preference were relatively more confidence in and relative ease of negotiating private health-care processes

10) #6 in the table on challenges--the role of the RST here seems like one the public health system should have taken on? this may warrant comment in the discussion

The RST only coordinated with the public health system to ensure supplies, which was needed only in the first three months of the study period, the actual procurement was done by the primary health facilities. The RST deliberately did not intervene in the procurement mechanisms, the initial coordination was done to help in streamlining the processes.

11) Figure 4: label the y-axis

Have labelled it as number of sick young infants.

Reviewer #2: 

Description of the study area has to be exhaustive. Number of staffs, their level of training and the referral linkages has to be clearly outlined. Coverage calculation has to be justified as per the comment given.

We have added a separate panel in figure no.3 that describes the health infrastructure. We have added coverage calculation in the first paragraph of the results section.

Supplemental tables are not attached or could not be downloaded. The whole manuscript needs to be checked according to the PLOS guideline. Font size, spacing,title and tables has to be revised.

We have submitted the supplementary tables and formatted the manuscript as per Plos One guidelines.

Additional Editor Comments (if provided):

This is the important study contributing towards the reduction in neonatal mortality in a resource-limited scenario. Authors have identified the service delivery outcome of the simplified management of PSBI among young infants as well as the implementation outcomes that is fidelity and feasibility.

In general, English language correction is deemed necessary.

In addition, authors should fulfil the following requirements to further strengthen the scientific rigour in reporting the implementation research.

1. To include key reference published in WHO Bulletin

Hales S, Lesher-Trevino A, Ford N et. al.

Reporting guidelines for implementation and operational research.

Bulletin of the World Health Organization 2016;94:58-64. doi: http://dx.doi.org/10.2471/BLT.15.167585

We have edited the manuscript for language corrections. We have tried to follow the mentioned guidelines as far as possible and have added this reference in the methods section. 

2. The authors needed to clarify the existing knowledge gaps for implementation barriers towards the service delivery guidelines in the Introduction section. What was the nature and severity of the problem?

We have added following text in the introduction section: 

However, implementation of such guideline in the public health system requires a multi-pronged approach and involves dialogue with the policy makers and program managers, understanding barriers and facilitators for implementation and providing technical support for implementation. While implementation challenges were well documented in controlled conditions in the above mention Asian and African countries, we could not find barriers to implementation of PSBI management within the public health system in India. Thus, an implementation research was planned across four different sites in India within the public health system to facilitate policy adoption and implementation of the WHO PSBI guideline. Maharashtra state has a relatively well functioning public health system and a private sector compared to many other states in India (20). However, its tribal areas still have poor access to health care due to lack of proper roads, inadequate number of health facilities as well as low level of income and education among tribal population (21). We describe here the process of implementation and outcomes of simplified management of PSBI in young infants in a tribal population from Western Maharashtra, India.

3. P. 6 LINES 119-120: This is a multisite and multi-country study. Please mention other countries supported by WHO to participate in this study to avoid competing interest in publication ethics.

Have added this: The other countries supported by WHO in this study were Democratic Republic of Congo, Ethiopia, Malawi, Nigeria and Pakistan.

4. To submit tools used for the baseline survey as additional files.

We have submitted tools used in baseline study used this study.

5. To add one Table to illustrate the results of the baseline survey clearly.

We have added a table showing the main findings as supplementary table number 4.

We have done accordingly.

2. Please address the following:

- Please ensure you have thoroughly discussed any potential limitations of this study within the Discussion section.

- Please include additional information regarding the survey used in the study and ensure that you have provided sufficient details that others could replicate the analyses. For instance, if you developed a survey as part of this study and it is not under a copyright more restrictive than CC-BY, please include a copy, in both the original language and English, as Supporting Information.

Instead of preparing a separate section on limitations, we have specifically written about limitations throughout the text of the discussion section.

We have taken note of this.

4. Please upload a copy of Supporting Information Table 1-2 which you refer to in your text on page 7 and 14.

We have uploaded a document on supplementary tables.

5. We note that Figure 2 in your submission contain map images which may be copyrighted. All PLOS content is published under the Creative Commons Attribution License (CC BY 4.0), which means that the manuscript, images, and Supporting Information files will be freely available online, and any third party is permitted to access, download, copy, distribute, and use these materials in any way, even commercially, with proper attribution. For these reasons, we cannot publish previously copyrighted maps or satellite images created using proprietary data, such as Google software (Google Maps, Street View, and Earth). For more information, see our copyright guidelines: http://journals.plos.org/plosone/s/licenses-and-copyright.

 1. You may seek permission from the original copyright holder of Figure 2 to publish the content specifically under the CC BY 4.0 license. 

We have edited the figure to include only a map that is copyrighted by KEM Hospital Research Centre Pune and have submitted a permission letter under CC BY 4.0 license.

---

## [Decision Letter · Decision Letter 1]

7 Jul 2020

Feasibility of implementation of simplified management of young infants with possible serious bacterial infection when referral is not feasible in tribal areas of Pune district, Maharashtra, India

PONE-D-20-03227R1

Dear Dr. Roy,

We’re pleased to inform you that your manuscript has been judged scientifically suitable for publication and will be formally accepted for publication once it meets all outstanding technical requirements.

Kind regards,

Khin Thet Wai, MBBS, MPH, MA (Population & Family Planning Resear

Academic Editor

PLOS ONE

Additional Editor Comments (optional):

Reviewers' comments:

Reviewer's Responses to Questions

**Comments to the Author**

1. If the authors have adequately addressed your comments raised in a previous round of review and you feel that this manuscript is now acceptable for publication, you may indicate that here to bypass the “Comments to the Author” section, enter your conflict of interest statement in the “Confidential to Editor” section, and submit your "Accept" recommendation.

Reviewer #1: All comments have been addressed

2. Is the manuscript technically sound, and do the data support the conclusions?

Reviewer #1: Yes

3. Has the statistical analysis been performed appropriately and rigorously? 

Reviewer #1: Yes

4. Have the authors made all data underlying the findings in their manuscript fully available?

Reviewer #1: Yes

5. Is the manuscript presented in an intelligible fashion and written in standard English?

Reviewer #1: Yes

6. Review Comments to the Author

Reviewer #1: The authors have addressed all of my concerns in my initial review. I have no additional comments and recommend accepting for publication.

7. PLOS authors have the option to publish the peer review history of their article (what does this mean?). If published, this will include your full peer review and any attached files.

Reviewer #1: No

---

## [Editor Report · Acceptance letter]

12 Aug 2020

PONE-D-20-03227R1 

Feasibility of implementation of simplified management of young infants with possible serious bacterial infection when referral is not feasible in tribal areas of Pune district, Maharashtra, India 

Dear Dr. Roy:

I'm pleased to inform you that your manuscript has been deemed suitable for publication in PLOS ONE. Congratulations! Your manuscript is now with our production department. 

Kind regards, 

on behalf of

Dr. Khin Thet Wai 

Academic Editor

PLOS ONE